# GPR97 triggers inflammatory processes in human neutrophils via a macromolecular complex upstream of PAR2 activation

Neutrophils play essential anti-microbial and inflammatory roles in host defense, however, their activities require tight regulation as dysfunction often leads to detrimental inflammatory and autoimmune diseases. Here we show that the adhesion molecule GPR97 allosterically activates CD177-associated membrane proteinase 3 (mPR3), and in conjugation with several protein interaction partners leads to neutrophil activation in humans. Crystallographic and deletion analysis of the GPR97 extracellular region identified two independent mPR3-binding domains. Mechanistically, the efficient binding and activation of mPR3 by GPR97 requires the macromolecular CD177/GPR97/PAR2/CD16b complex and induces the activation of PAR2, a G protein-coupled receptor known for its function in inflammation. Triggering PAR2 by the upstream complex leads to strong inflammatory activation, prompting anti-microbial activities and endothelial dysfunction. The role of the complex in pathologic inflammation is underscored by the finding that both GPR97 and mPR3 are upregulated on the surface of disease-associated neutrophils. In summary, we identify a PAR2 activation mechanism that directs neutrophil activation, and thus inflammation. The PR3/CD177/GPR97/PAR2/CD16b protein complex, therefore, represents a potential therapeutic target for neutrophil-mediated inflammatory diseases.

Neutrophils express a plethora of immune effectors including proteases, cytokines, and receptors which contribute significantly to both the innate and adaptive immune responses[1]. However, these neutrophil-derived molecules require stringent regulation as their uncontrolled activities are often associated with detrimental inflammatory and autoimmune diseases[2]. Neutrophil serine proteases (NSPs) are azurophilic granule proteins which play critical roles in anti-microbial and inflammatory responses[3,4]. Proteinase 3 (PR3) is unique among NSPs as it also represents the designated auto-antigen of granulomatosis with polyangiitis (GPA), an autoimmune disorder characterised by the inflammatory necrosis of small/medium-sized blood vessels and the common presence of anti-neutrophil cytoplasmic antibodies (ANCA) against PR3[5]. Interestingly, a fraction of PR3 is exocytosed and tethered to neutrophil membrane (termed mPR3)

mainly by specific binding to the glycosylphosphatidylinositol-anchored CD177[6]. Although increased plasma PR3 and mPR3 levels are positively associated with GPA pathogenesis, the normal cellular function of mPR3 remains enigmatic[7,8].

G protein-coupled receptors (GPCRs) are critically involved in the activation and chemotaxis of neutrophils, and deregulated GPCR activities in neutrophils usually result in immune dysfunction and clinical disease[9]. Protease-activated receptors (PARs) represent a group of unique GPCRs that are activated by a tethered peptide ligand exposed by the specific proteolytic processing of their N-terminal region[10]. Activation of PAR2, the predominant PAR expressed by neutrophils, is involved in tissue inflammation and prenatal death associated with the autoimmune antiphospholipid syndrome[11,12]. Although trypsin is the prototypic activator of PAR2, many other proteinases are

✉ e-mail: elena.seiradake@bioch.ox.ac.uk; hhlin@mail.cgu.edu.tw

similarly able to activate or disarm PAR2[13]. Intriguingly, PAR2 can also be transactivated by PAR1 by the formation of PAR1-PAR2 heterodimers[14]. To date, it is not fully understood how PAR2 activation is triggered in human neutrophils and whether PAR2 transactivation occurs in neutrophils.

Adhesion GPCRs (aGPCRs) are atypical GPCRs, having a large extracellular region (ECR) before the seven-transmembrane domain[15]. Most aGPCRs are expressed as a non-covalent bipartite complex due to the auto-proteolytic cleavage of ECR at the GPCR proteolysis site (GPS) within the GPCR autoproteolysis-inducing (GAIN) domain[15]. Consequently, many aGPCRs are activated by a tethered agonism mechanism remarkably similar to that of PARs which involves the unmasking of an internal agonistic peptide following the dissociation/dislocation of ECR upon binding to its cellular ligand(s)[16]. Human neutrophils express several aGPCRs including EMR2/ADGRE2, EMR3/ADGRE3, CD97/ADGRE5, and GPR97/ADGRG3[17,18]. Although an anti-microbial role has been demonstrated for GPR97 previously, the lack of specific ligands has hindered the mechanistic understanding of its function.

Here we show that GPR97 is the binding partner and allosteric activator of mPR3 through the macromolecular CD177/GPR97/PAR2/CD16b receptor complex. Intriguingly, GPR97-augmented mPR3 enzymatic activity cleaves and activates PAR2 leading to robust neutrophil activation. In conclusion, our results uncover an aGPCR-GPCR activation mechanism in human neutrophils that contributes to inflammatory responses.

## Results

### Activated neutrophils up-regulate GPR97 expression which correlates with the inflammation and/or disease status of various inflammatory disorders

Up-regulated GPR97 transcript and protein have been identified in blood neutrophils and tissue-infiltrating neutrophils of diverse inflammatory disorders[18]. To further confirm the relationship of neutrophil GPR97 expression and tissue inflammatory status, we compared GPR97 expression in early- and late-stage appendicitis. Specific GPR97 expression was consistently detected in tissue-infiltrating neutrophils which were more numerous in late-stage appendicitis (Fig. 1a, b). Moreover, much higher GPR97 staining intensities were identified in the late-stage than the early-stage appendicitis (Fig. 1c).

Similarly, higher surface GPR97 levels were detected in blood neutrophils of bacterial sepsis patients than those of healthy controls by flow cytometry analyses (Fig. 1d). Thereafter, we enrolled patients of GPA and microscopic polyangiitis (MPA), two distinct autoimmune diseases characterized by ANCA mostly directed toward PR3 and myeloperoxidase (MPO), respectively[5]. No significant differences in GPR97 expression were observed among healthy controls and patients (Fig. 1e). However, when patients were divided into remission and active disease groups, much higher GPR97 levels were detected in neutrophils of patients with signs of active disease (Fig. 1f). Moreover, GPR97 expression levels also correlated positively to the ANCA titers of patients (Fig. 1g). Altogether, up-regulated GPR97 expression in activated neutrophils is positively associated with disease activity of various inflammatory disorders.

### Neutrophils are activated by GPR97 binding to a putative ligand expressed exclusively in a human neutrophil subpopulation

Given that many aGPCRs have cell-associated ligands, a panel of cell lines and primary cells were screened via flow cytometry for potential GPR97 ligands[19]. A chimeric GPR97E-mFc protein containing the human GPR97-ECR fused with a mouse fragment crystallizable (Fc) was used as the probe (Fig. 2a, and Supplementary Fig. 1a). From over 30 different cell samples screened, a putative GPR97-ligand was successfully identified which was expressed in a unique bimodal pattern exclusively on human neutrophil surfaces (Fig. 2b, c, and Supplementary Fig. 1b).

Intriguingly, the GPR97-ligand+ neutrophil subpopulation ranged from 40–90% among different donors but remained rather constant in the same individuals over time (Fig. 2d). Ligand-binding was not detected in human neutrophils probed by mouse GPR97E-mFc, nor in mouse neutrophils probed by human or mouse GPR97E-mFc probes (Supplementary Fig. 1c). Hence, the interaction of GPR97 and its putative ligand is a human neutrophil-specific feature and the expression of GPR97-ligand is strictly regulated in neutrophils.

Importantly, incubation of resting neutrophils with GPR97E-mFc triggered robust activation, resulting in several distinct phenotypic manifestations. These include cell shape changes, enhanced production of reactive oxygen species (ROS) and interleukin-8 (IL-8), increased MPO activity, as well as expressional changes of multiple neutrophil activation markers such as CD62L, Mac-1, CD54, and CXCR1 (Fig. 2e–j, and Supplementary Fig. 1d–g). Notably, IL-8 production was up-regulated comparably by neutrophils treated with soluble or immobilized GPR97E-mFc in a dose- and time-dependent manner (Fig. 2j, and Supplementary Fig. 1h). In summary, a putative GPR97-specific ligand is identified exclusively in a distinct human neutrophil subset whose interaction with GPR97E-mFc induces inflammatory activation.

### GPR97 is a specific binding partner and allosteric activator of neutrophil mPR3

The known ligands of aGPCRs include lipids, glycosaminoglycans, and proteins[15]. To delineate the putative GPR97-ligand, far-western blot analysis of neutrophil proteins was performed using GPR97E-mFc as a probe. Three distinct signals were detected specifically in the membrane fraction, including a prominent ~29 kDa protein and two weaker signals of ~52 and ~140 kDa proteins (Fig. 3a), suggesting that the GPR97-ligand is likely a membrane protein. To date, CD177 (~50–60 kDa), PR3 (~29 kDa), and olfactomedin 4 (OLFM4)(~57 kDa) are all known to show a bimodal profile in human neutrophils hence are strong GPR97-ligand candidates[20]. The glycosylphosphatidylinositol-linked CD177 is a neutrophil-restricted Ly6 receptor family member expressed in a characteristic 0–100% pattern in different individuals due to unique genetic mechanisms[21,22]. The intracellular elastase-like PR3 is somewhat exocytosed and forms mPR3 on resting neutrophils mainly by binding to CD177, thus displaying a CD177-like expression pattern[6]. By contrast, OLFM4 is a specific granule resident protein which is secreted upon neutrophil activation and is not known to tether on the neutrophil membrane[23].

Interestingly, the putative GPR97-ligand was sensitive to digestion by phosphoinositide phospholipase C that cleaves the glycosylphosphatidylinositol-linker (Fig. 3b, and Supplementary Fig. 2a). Moreover, the GPR97-ligand was detected exclusively in the CD177+ and mPR3+ neutrophil subsets, showing closely matched expression profiles and strong co-localization with CD177 and mPR3 (Fig. 3c, and Supplementary Fig. 2b). By contrast, only 20–40% of neutrophils expressing OLFM4 or glycosylphosphatidylinositol-linked CD55 were GPR97-ligand+ (Supplementary Fig. 2c). Finally, morphological changes and IL-8 up-regulation induced by GPR97E-mFc were detected solely in the CD177+ but not CD177− neutrophil subset (Fig. 3d, e, and Supplementary Fig. 2d). These results suggested strongly a direct interaction of GPR97E-mFc with either CD177 or mPR3. Thereafter, far-western blot analyses revealed specific binding of GPR97E-mFc to PR3 but not CD177 (Fig. 3f). We subsequently verified the specific GPR97-PR3 interaction using two different PR3-binding assays, namely the FACS-based ligand-binding assay performed in CD177- and GPR97-expressing HEK-293T cells and the enzyme-linked immunosorbent assay (ELISA)-like protein-protein binding assay (Fig. 3g–i, and Supplementary Fig. 2e, f). Both assays showed that GPR97, like CD177, binds to PR3 specifically, albeit much less efficiently. Altogether, we conclude that GPR97-ECR interacts specifically with the PR3 moiety of the mPR3 (PR3-CD177) complex on human neutrophil surfaces.

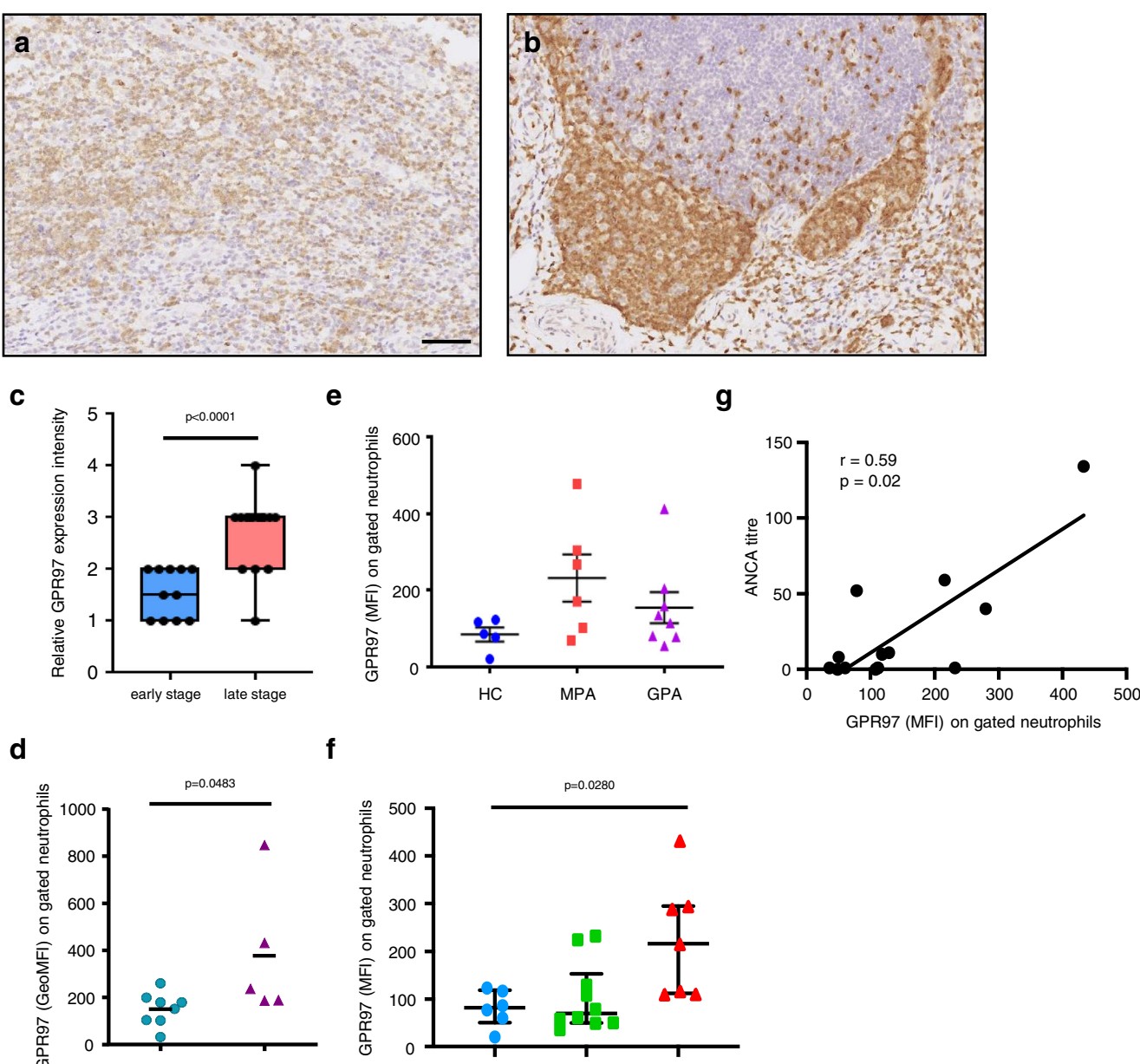

**Fig. 1 | Up-regulated GPR97 expression in neutrophils correlates with the disease status of inflammatory disorders. a, b** The immunohistochemical analyses of GPR97 expression in tissue sections of the early- **a** and late-stage **b** appendicitis. Reactivity of the anti-GPR97 mAb to GPR97 was shown as brownish staining. Scale bar, 100 μm. **c** The relative GPR97 staining intensities in tissue-infiltrating neutrophils of the early- ($n = 11$) and late-stage ($n = 16$) appendicitis tissues. Data are means ± SEM and $p$ value was determined by one-sided unpaired student's t-test. **d** Flow cytometry analyses of surface GPR97 levels of gated neutrophils of HC ($n = 8$) and bacterial sepsis patients ($n = 5$). Data are means ± SEM and $p$ value was determined by two-sided unpaired student's t-test. **e, f** Flow cytometry analyses of surface GPR97 levels of gated neutrophils of HC ($n = 5$ in **e**, 6 in **f**) and patients based on the disease category (GPA $n = 8$, MPA $n = 6$) **e** and status (remission $n = 10$, active $n = 7$) **f. g** The positive correlation of GPR97 levels with the ANCA titers of GPA ($n = 8$) and MPA ($n = 6$) patients. Data are means ± SEM and $p$ value was determined by one-way ANOVA. HC healthy control, GPA granulomatosis with polyangiitis, MPA microscopic polyangiitis. Source data are provided in the Source Data file.

The enzymatic activity of mPR3 is much reduced compared to soluble PR3 due to the PR3-CD177 interaction[24]. This prompted us to investigate the potential role of mPR3 activity in GPR97-induced neutrophil activation and the possible effect of GPR97-mPR3 binding to its proteolytic activity. Surprisingly, GPR97-induced neutrophil activation was significantly attenuated in the presence of broad-spectrum serine protease inhibitors including α1-antitrypsin (A1AT), aprotinin, and TPCK, but not the cysteine protease inhibitor E-64 (Fig. 3j, and Supplementary Fig. 3a). Accordingly, two endogenous NSP inhibitors, elafin and secretory leukocyte peptidase inhibitor (SLPI), were employed next. SLPI only inhibits neutrophil elastase (NE) and cathepsin G (CG), while elafin inhibits NE, CG, and PR3[25].

Interestingly, IL-8 up-regulation was dramatically inhibited in elafin- but not SLPI-treated neutrophils (Fig. 3k). These findings established that mPR3 protease activity is essential for GPR97-induced neutrophil activation and implied that GPR97 binding to mPR3 likely promotes its proteolytic activity. We therefore determined the ex vivo enzymatic activity of mPR3 using a PR3-specific FRET-based substrate[26]. As expected, incubation of neutrophils with GPR97^E-mFc resulted in augmented mPR3 activities which were inhibited by A1AT and elafin, but not SLPI without apparent degranulation of azurophilic granules (Fig. 3l, m, and Supplementary Fig. 3b–d). Thus, we conclude that GPR97 is a binding partner and allosteric activator of mPR3.

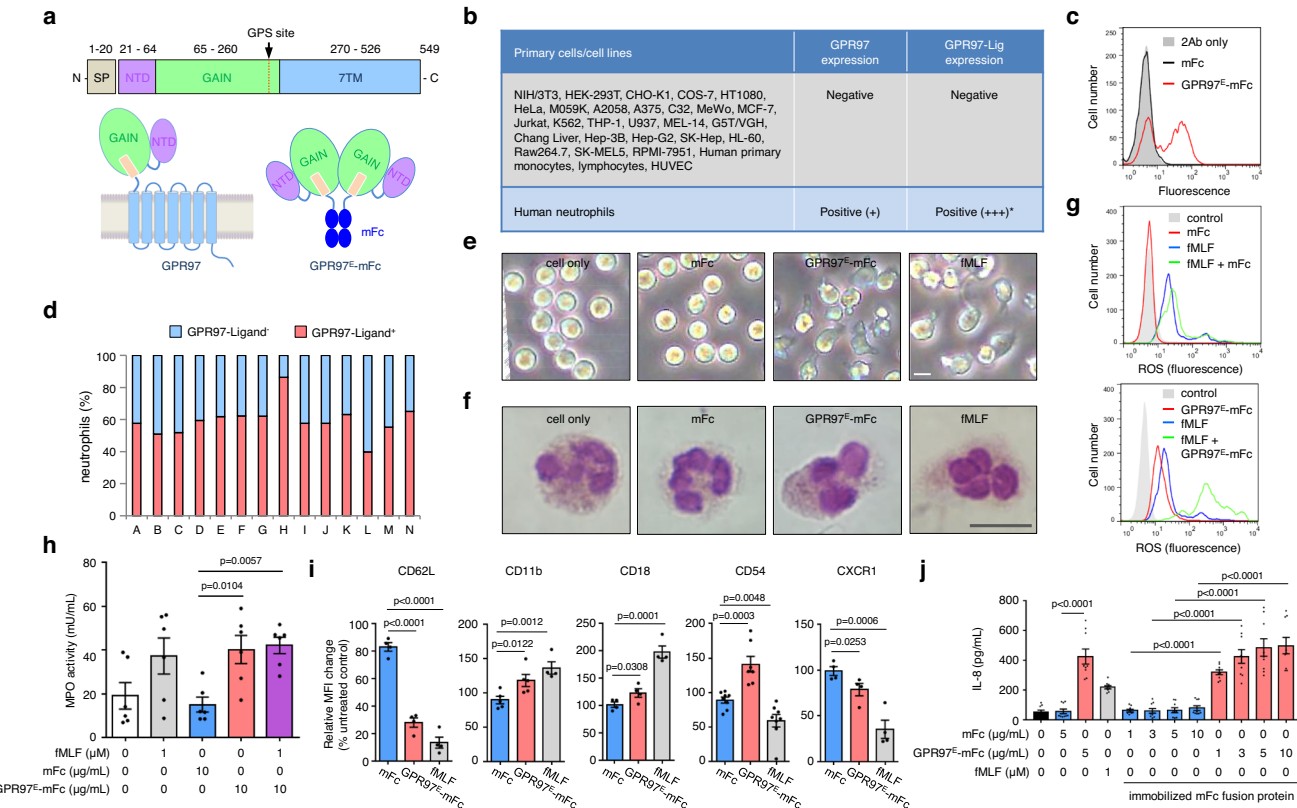

**Fig. 2 | Identification of a putative GPR97-ligand in a distinct human neutrophil subpopulation that is activated upon interaction with GPR97E-mFc.**
**a** Schematics of GPR97 receptor and GPR97E-mFc probe. **b** List of cell samples screened for the expression of GPR97 and the putative GPR97-ligand. * indicates a subpopulation of primary human neutrophils. **c** A representative bimodal surface expression profile of the GPR97-ligand in human neutrophils detected by the GPR97E-mFc probe. **d** Highly variable GPR97-ligand⁻ (blue) and GPR97-ligand⁺ (pink) neutrophil subsets in bloods of different donors (samples A to N).
**e, f** Morphological changes of neutrophils induced by GPR97E-mFc as observed by inverted light microscope **e** and Wright-Giemsa stain **f**. Scale bar, 20 µm. Experiments were repeated 3 times with similar results. **g–j** Phenotypic analyses of activated neutrophils induced by GPR97E-mFc included up-regulated ROS production **g**, MPO activity (**h**, $n = 6$), expressional changes of specific CD markers (**i**, $n = 7$ for CD54 and $n = 4$ for others) and IL-8 production (**j**, $n = 3$) of independent samples. The mFc protein and fMLF were used as a negative and a positive control, respectively. Data are presented as means ± SEM and p value was determined by two-way ANOVA in **h**, **j** and one-way ANOVA in **i**. Source data are provided in the Source Data file.

## Atomic resolution of the GPR97 extracellular region and mapping of the mPR3-binding domains

The ECR of most aGPCRs contains an extended N-terminal segment followed by a GAIN domain both of which are able to interact with distinct ligands[15]. Previous structural studies have shown that the canonical GAIN domain is evolutionarily conserved and includes A and B subdomains[27]. Subdomain A typically consists of 6 α-helices, while subdomain B comprises a twisted β-sandwich made of 13 β-strands and 2 small α-helices[27]. Nevertheless, structural analyses of GPR56/ADGRG1 and GPR126/ADGRG6 have revealed a smaller subdomain A with fewer α-helices[28,29]. Interestingly, our bioinformatic analyses predicted that GPR97 contains an even smaller GAIN domain than GPR56 and GPR126.

To gain a better insight into the structural organization and mPR3-binding characteristics of GPR97-ECR, we determined its atomic structure (residues 28–260) at 3.37 Å resolution using X-ray crystallography (Fig. 4a–c, and Supplementary Fig. 4a–e). The structural analyses showed the GPR97-GAIN domain retains a complete subdomain B but contains only one α-helix in subdomain A, approximately where α6 of the canonical GAIN-subdomain A would be located (Supplementary Fig. 4f). The electron density map also confirmed that the GPR97 GAIN domain is mostly auto-proteolysed at its GPS motif, as expected and observed for other GAIN domains (Supplementary Fig. 1a)[18,27–29]. Interestingly, we identified a small N-terminal domain (NTD) adjacent to the GAIN domain, which contains four cysteine residues (C32, C50, C54, and C62) that are highly conserved among GPR97 orthologs (Supplementary Fig. 5a). Sulphur-SAD phasing was hence used to precisely locate

the cysteine residues and the resulting model shows that the single helix of the subdomain A (helix α6 in canonical GAIN domains) packs against subdomain B and acts as an anchor for the upstream helix of NTD. A cysteine bridge stabilises the packing of the NTD helix against the subdomain A helix (C54-C62) (Fig. 4a, b). The NTD helix is further stabilised by extensive hydrophobic interactions with the GAIN domain, involving the residues Y39, L44, F51 from the NTD and L67, Y70, W71, Y74, H77 and F91 from the GAIN domain, and a cysteine bridge between the helix and the N-terminal loop of the NTD (C32-C50) (Fig. 4c). These hydrophobic residues are highly conserved specifically among GPR97 orthologs (Supplementary Fig. 5a,b). Structural comparison with a model calculated by alpha-fold[30] shows that the GAIN domain is similar, but the NTD-helix is off-set compared to our experimental model. Alpha-fold predicts residues Q21-G27 to be flexible (Supplementary Fig. 4i). Taken together, the GPR97-ECR consists of a small cysteine-stabilised helical NTD followed by an unusually small GAIN domain that consists of a subdomain A with one α-helix and a typical subdomain B with 13 β-strands (Fig. 4a).

To map the mPR3-binding region(s), we generated domain-swapped mFc-fusion probes by interchanging the GAIN domains of GPR97 and GPR56, two closely related ADGRG subfamily members (Fig. 4d, and Supplementary Fig. 6a). A similar ligand-binding signal was detected by the GPR97NTD/GPR56GAIN-mFc and GPR56PLL/GPR97GAIN-mFc probes as did GPR97E-mFc, while GPR56E-mFc showed nil binding (Fig. 4d). These results suggested that both the GPR97-NTD and -GAIN domains contain one independent mPR3-binding region, respectively.

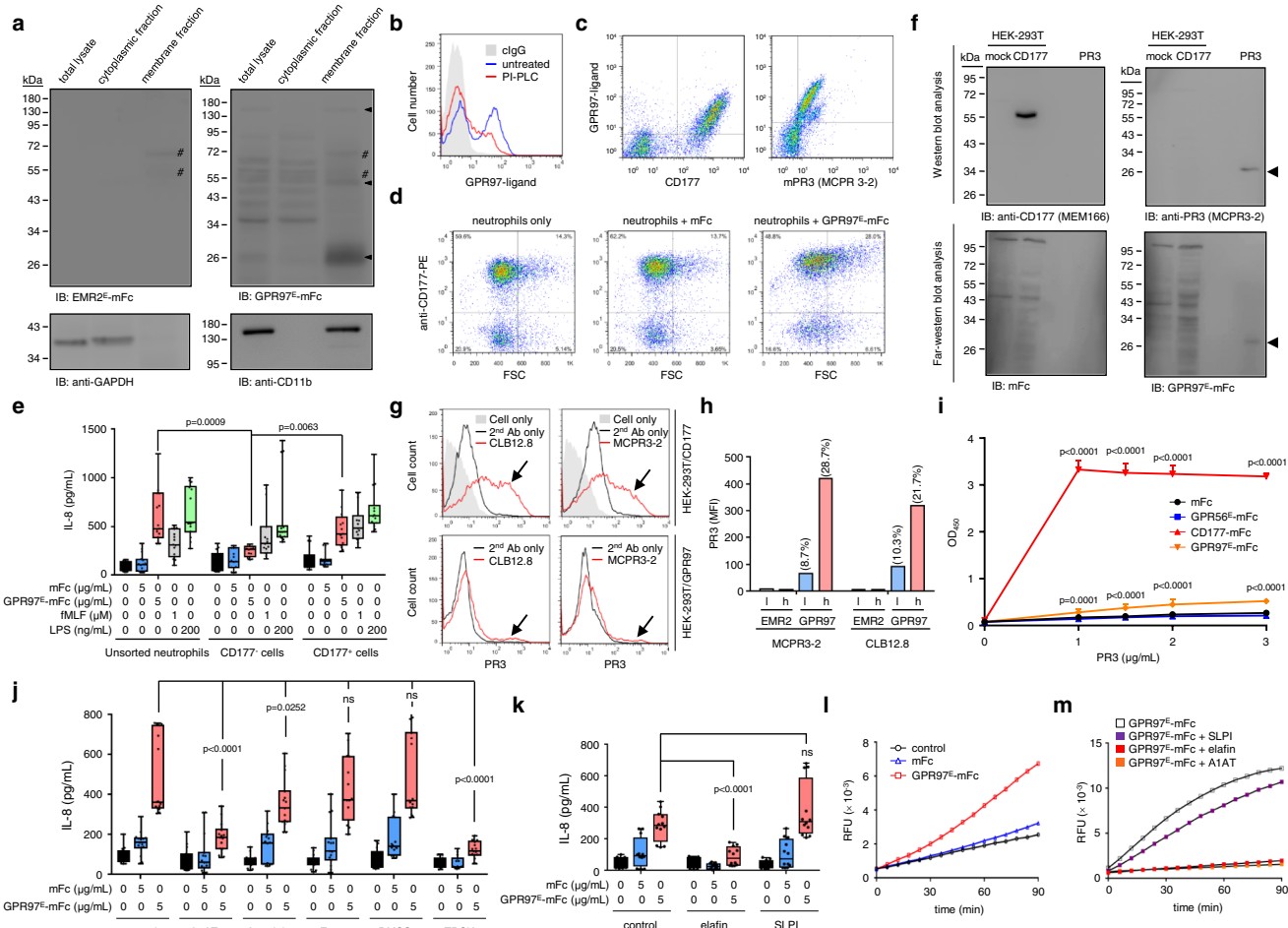

**Fig. 3 | GPR97 is a binding partner and allosteric activator of mPR3. a** Far-western blot analyses of the putative GPR97-ligand(s)(black arrowheads). The EMR2$^E$-mFc protein was used as a negative control. Anti-CD11b and anti-GADPH Abs detected the corresponding protein markers of the membranous and cytoplasmic fractions, respectively. #, non-specific signals. Experiments were repeated 3 times with similar results. **b** FACS-based ligand-binding analysis of neutrophils pre-treated without or with phosphoinositide phospholipase C (PI-PLC). **c** Dot-plots of neutrophils double-stained with GPR97$^E$-mFc and anti-CD177 or anti-PR3 mAb. **d** Dot-plots (FSC vs. surface CD177 levels) of neutrophils incubated with GPR97$^E$-mFc. **e** IL-8 produced by unsorted and sorted neutrophils incubated with various reagents as indicated. $n = 5$ independent experiments. Cells treated with fMLF and LPS were included as positive controls. **f** Far-western blot analysis of specific GPR97-PR3 binding. Samples included total lysates of mock- and CD177-transfected HEK-293T cells as well as purified PR3. The arrowheads indicated the specific signals

detected by the anti-PR3 mAb and the GPR97$^E$-mFc probe. Experiments were repeated 3 times with similar results. **g**, **h** The FACS-based PR3-binding assay of unsorted **g** and FACS-sorted **h** HEK-293T cells expressing CD177 or GPR97. The black arrows in **g** indicated the specific PR3-binding. FACS-sorted GPR97$^{high}$ **h** and GPR97$^{low}$ **l** HEK-293T cells were examined. Numbers on top of the bars indicated the percentage of PR3-binding cell populations. **i** ELISA-like PR3-binding analyses. CD177-mFc and GPR56-mFc proteins were included as the positive and negative controls, respectively. $n = 6$ biologically independent samples. **j**, **k** ELISA analysis of IL-8 secreted by neutrophils treated without or with indicated protease inhibitors. $n = 5$ **j**, 4 **k** independent experiments. Data are presented as means ± SEM and $p$ value was determined by two-way ANOVA in **e**, **i**–**k**. ns non-significant. **l**, **m** The ex vivo mPR3 enzymatic activity of neutrophils incubated in the absence or presence of mFc or GPR97-mFc protein probe without or with protease inhibitors as indicated. Source data are provided in the Source Data file.

Domain-truncated GPR97-mFc probes were used subsequently to confirm the two different domains can indeed bind to mPR3 individually (Fig. 4e). Unexpectedly however, these domain-swapped and -truncated mFc-fusion proteins induced much weaker neutrophil activation phenotypes compared to the full-length GPR97$^E$-mFc (Fig. 4f, g, and Supplementary Fig. 6b). Similarly, only GPR97$^{GAIN}$-mFc among the various GPR97-mFc probes stimulated a moderately-increased mPR3 activity (Fig. 4h). We conclude that the maximal GPR97-induced neutrophil activation is achieved when mPR3 is bound simultaneously by the two ligand-binding regions in the GPR97-NTD and -GAIN domains.

## GPR97-mPR3 interaction requires a macromolecular CD177/GPR97/PAR2/CD16b receptor complex and triggers PAR2 activation in neutrophils

As CD177 is incapable of direct signaling due to the lack of a transmembrane moiety, the results above implicated strongly the

involvement of an undefined signaling molecule triggered by the GPR97-augmented mPR3 in inducing neutrophil activation. In order to identify this signal transducer, a heterologous HEK-293T cell transient expression system was established. Unexpectedly, while exogenous PR3 bound readily to CD177-expressing HEK-293T cells hence forming mPR3, no GPR97$^E$-mFc binding was ever detected in these cells (Fig. 5a). We thus speculated that additional co-receptor(s) expressed in neutrophils but not HEK-293T cells is needed for efficient GPR97-mPR3 binding and likely represents the signaling molecule of interest.

To date, several interacting proteins/substrates of CD177 and PR3, including the endothelial protein C receptor (EPCR), CD16b (FcγRIIIb), Mac-1 (CD11b/CD18), and PAR2 have been identified in neutrophils[31–34]. Hence, we co-expressed CD177 and these proteins in different combinations in HEK-293T cells. Surprisingly, a positive GPR97-mPR3 binding signal was only detected in cells co-expressing the specific combination of CD177, GPR97, PAR2 and CD16b (Fig. 5a, b). Moreover,

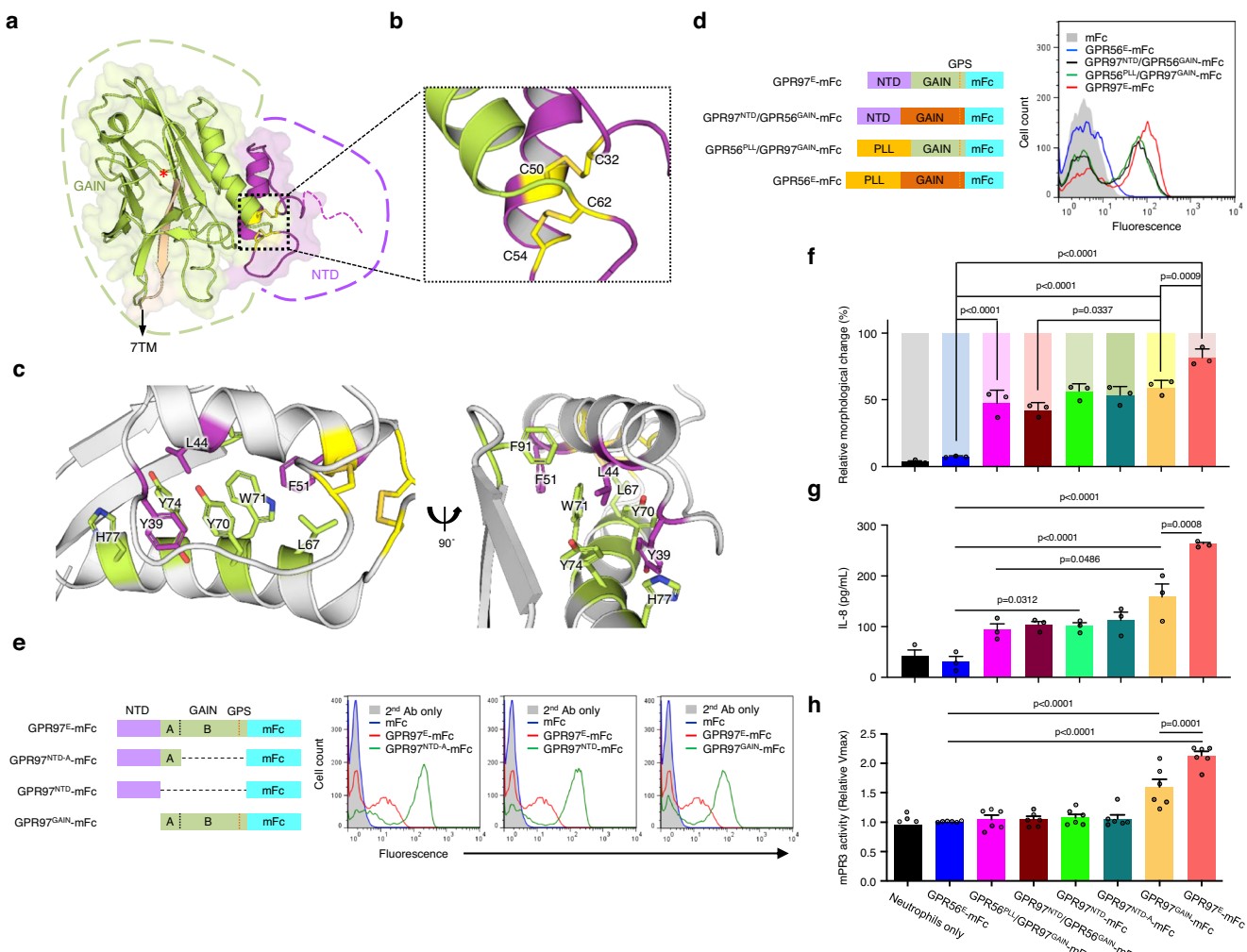

**Fig. 4 | Structural analysis of GPR97-ECR and mapping of the mPR3-binding regions. a** Cartoon representation of the GPR97-ECR structure obtained from X-ray crystallography experiments. The NTD is depicted in purple and the GAIN domain in green and beige (the 13th β-strand). Cysteine residues involved in linking and stabilising the NTD are shown as yellow sticks. The red asterisk indicates the GPS cleavage site. **b** Close-up of the two disulphide bridges, C32-C50 and C54-C62, that are involved in the stabilisation of the NTD and GAIN domains. **c** Close-up on the extensive hydrophobic interactions stabilising the packing of the NTD against the GAIN domain. Residues from the NTD are depicted in purple and those from the GAIN domain are depicted in green. **d**, **e** Mapping of the mPR3-binding region(s)

using the domain-swapped **d** and domain-truncated **e** GPR97-mFc probes as indicated. *n* = 5 **d** and 6 **e** independent experiments. **f**–**h**. Activated neutrophil phenotypes, namely morphological changes **f**, IL-8 production **g**, and the ex vivo mPR3 enzymatic activity **h** treated with various GPR97-mFc probes as indicated. *n* = 3 **f**, 4 **g**, and 6 **h** independent experiments. The bright and dim colors in **f** represent the percentage of irregular-shaped and round-shaped neutrophils, respectively. The results shown in **h** are relative Vmax of the ex vivo mPR3 enzymatic activity. Data are presented as means ± SEM and *p* value was determined by two-way ANOVA. Source data are provided in the Source Data file.

the mPR3-binding signal was lost when any member of the four-receptor complex was replaced with a different protein of the same receptor type, indicating that every co-receptor is indispensable for efficient GPR97-mPR3 binding (Fig. 5c). Next, the proximity-ligation assay (PLA) was carried out to verify the close association of various receptor-pairs of the CD177/GPR97/PAR2/CD16b complex on the neutrophil membrane (Fig. 5d, and Supplementary Fig. 3e). These results indicate that a clustered CD177/GPR97/PAR2/CD16b receptor complex is minimally required for efficient GPR97-mPR3 binding.

PAR2 is the predominant PAR expressed by human neutrophils and its activation is known to induce activation phenotypes similar to those triggered by GPR97E-mFc[12]. PAR2 hence seemed the most likely signaling transducer activated by GPR97-augmented mPR3. Indeed, the surface PAR2 levels of both neutrophils and PR3-bound HEK-293T transfectants were significantly reduced when incubated with GPR97E-mFc (Fig. 5e, f). Most critically, enhanced IL-8 production by GPR97E-mFc-treated neutrophils was significantly and specifically attenuated in the presence of a functional-blocking anti-PAR2 mAb and PAR2-

specific antagonists, while the PAR1 antagonist had no inhibitory effect (Fig. 5g). These results clearly demonstrated that PAR2 is not only the essential component of the CD177-associated complex, but also is the signaling receptor targeted by the GPR97-augmented mPR3. Thus, we propose a GPR97-PAR2 activation mechanism that involves PR3 and the CD177/GPR97/PAR2/CD16b complex. More precisely, GPR97 binds specifically to mPR3 presented by the CD177-associated receptor complex and promotes its enzymatic activity which in turn cleaves and activates PAR2 (Fig. 5h).

## Up-regulated GPR97 and PAR2 expression induces GPR97-PAR2 activation in activated neutrophils

Resting human neutrophils are generally inert suggesting no competent induction of GPR97-PAR2 activation which is almost certainly regulated by the expression levels of individual members of the CD177-associated receptor complex. Indeed, our results showed that while CD16b, mPR3, and CD177 were highly expressed in all or some (for mPR3 and CD177) resting neutrophils, GPR97 and PAR2 were weakly,

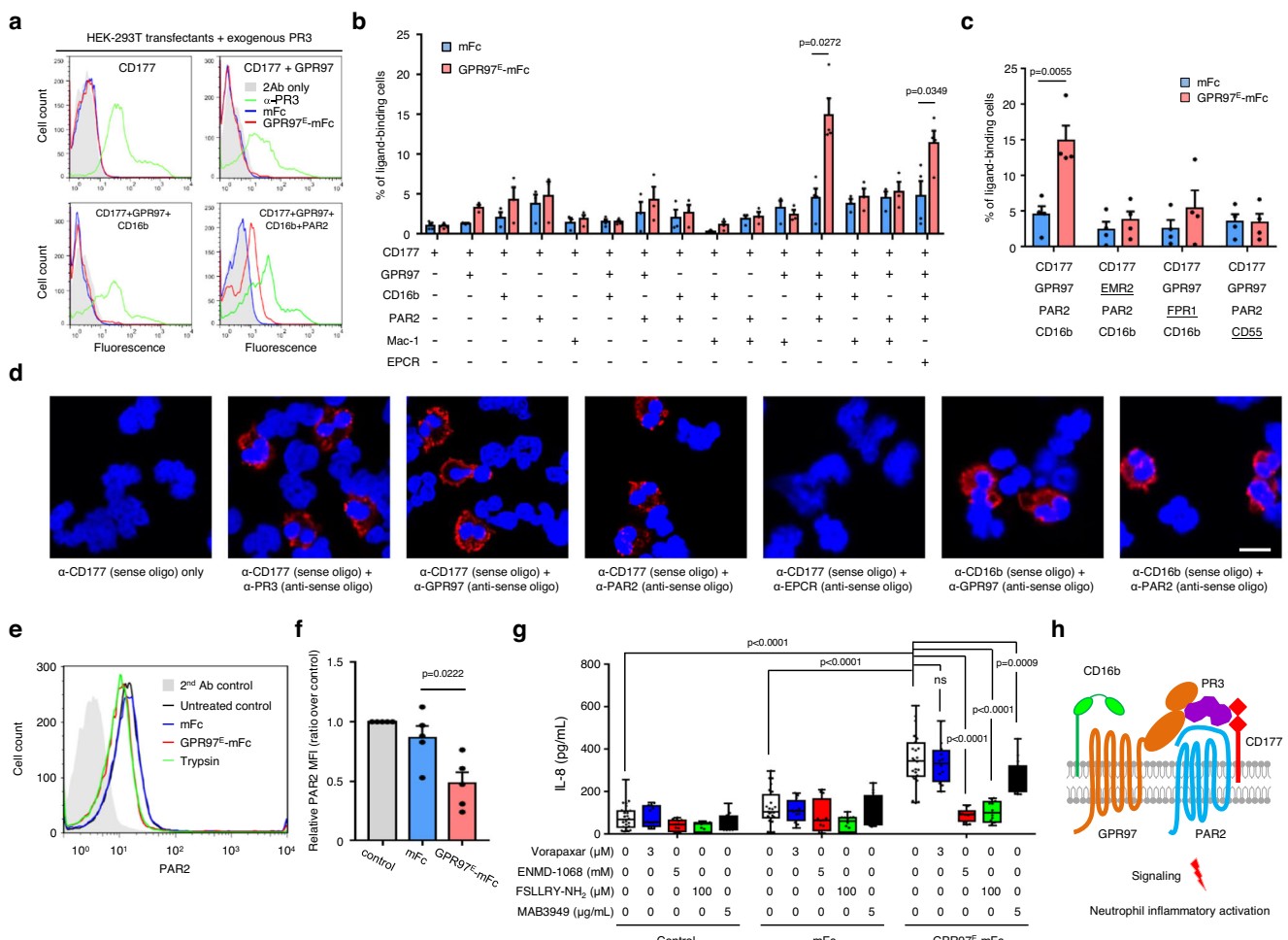

**Fig. 5 | The CD177/GPR97/PAR2/CD16b receptor complex is required for efficient GPR97-mPR3 interaction that induces PAR2 activation. a–c** Flow cytometry analyses of GPR97-mPR3 interaction in HEK-293T cells expressing various receptors as indicated. **a** Surface levels of membrane-bound PR3 (green), GPR97$^E$-mFc (red), and mFc (blue) were detected using appropriate Abs. **b**, **c** The percentage of transfectants displaying the positive GPR97$^E$-mFc binding signal in the ligand-binding assay. $n = 3$ **b**, 4 **c** independent experiments. The underlined receptor in **c** indicates the substituted component of the core receptor complex. Data are means ± SEM and $p$ value was determined by two-sided unpaired student's t-test. **d** The PLA analyses showed the interaction signals of specific receptor pairs of the CD177-associated complex in resting neutrophils. Scale bar, 10 μm. Experiments were repeated 3 times with similar results. **e**, **f** Flow cytometry analyses of PAR2 proteolysis in neutrophils **e** and HEK-293T cells expressing the CD177-associated receptor complex **f**. Trypsin-treated neutrophils and mFc were included as a positive and a negative control, respectively. $n = 5$ independent experiments. Data are means ± SEM and $p$ value was determined by one-way ANOVA. **g** ELISA analyses of IL-8 secreted by neutrophils incubated with the indicated reagents. $n = 4$ independent experiments. Data are means ± SEM and $p$ value was determined by two-way ANOVA. ns, non-significant. **h** Schematic diagram of the PR3/CD177/GPR97/PAR2/CD16b complex in neutrophils. CD16, a green-colored receptor with two connected ovals; GPR97, a brown-colored 7TM receptor with two extracellular ovals; PAR2, a blue-colored 7TM receptor; PR3, a purple-colored soluble molecule; CD177, a red-colored receptor with two diamonds. Source data are provided in the Source Data file.

sometimes barely, expressed (Fig. 6a, and Supplementary Fig. 7a). Interestingly, we detected increased PAR2, but not GPR97 expression only in neutrophils treated with certain inflammatory stimulants such as IFN-γ, indicating a stringent expressional regulation of GPR97 and PAR2 expression (Supplementary Fig. 7b, c). It is likely that low expression levels of GPR97 and PAR2 limit the initiation of GPR97-PAR2 activation and the spontaneous self-activation of resting neutrophils.

In line with previous results, significant up-regulation of both GPR97 and PAR2 was identified when resting neutrophils were stimulated by degranulation stimulants of azurophilic granules or by heat-aggregated IgGs (aIgGs) via an Fc receptor (FcR)-dependent mechanism (Fig. 6a, b). Consequently, PLA analyses revealed significantly increased interactions between receptor pairs of the CD177-associated complex in aIgG-activated neutrophils, compared to resting controls (Fig. 6c). Importantly, less IL-8 was produced in aIgG-stimulated neutrophils in the presence of FcR blocker, serine protease inhibitors,

PAR2 antagonists, or functional blocking anti-PAR2 and anti-GPR97 mAbs, suggesting a requirement for GPR97-PAR2 activation (Fig. 6d).

To evaluate the clinical relevance of FcR-induced GPR97-PAR2 activation, neutrophils were incubated with purified PR3-ANCA IgGs which activate neutrophils via FcγRs[35]. Indeed, increased GPR97 and PAR2 expression was detected in resting neutrophils incubated with PR3-ANCA, but not MPO-ANCA IgGs, in a FcR-dependent manner (Supplementary Fig. 8a, b). Critically, PR3-ANCA IgGs caused extensive neutrophil aggregates and led to increased production of IL-8. This effect was also attenuated in the presence of FcR blocker, serine protease inhibitors, or functional blocking anti-PAR2 and anti-GPR97 mAbs (Fig. 6e, f, and Supplementary Fig. 8c, d). Altogether, we conclude that most PAR2 and GPR97 are stored in azurophilic granules and translocated to cell surfaces when neutrophils received specific activation signals. The increased presence of PAR2 and GPR97 at the cell surface promotes clustering of CD177-associated receptors and induces GPR97-PAR2 activation, leading to further neutrophil activation.

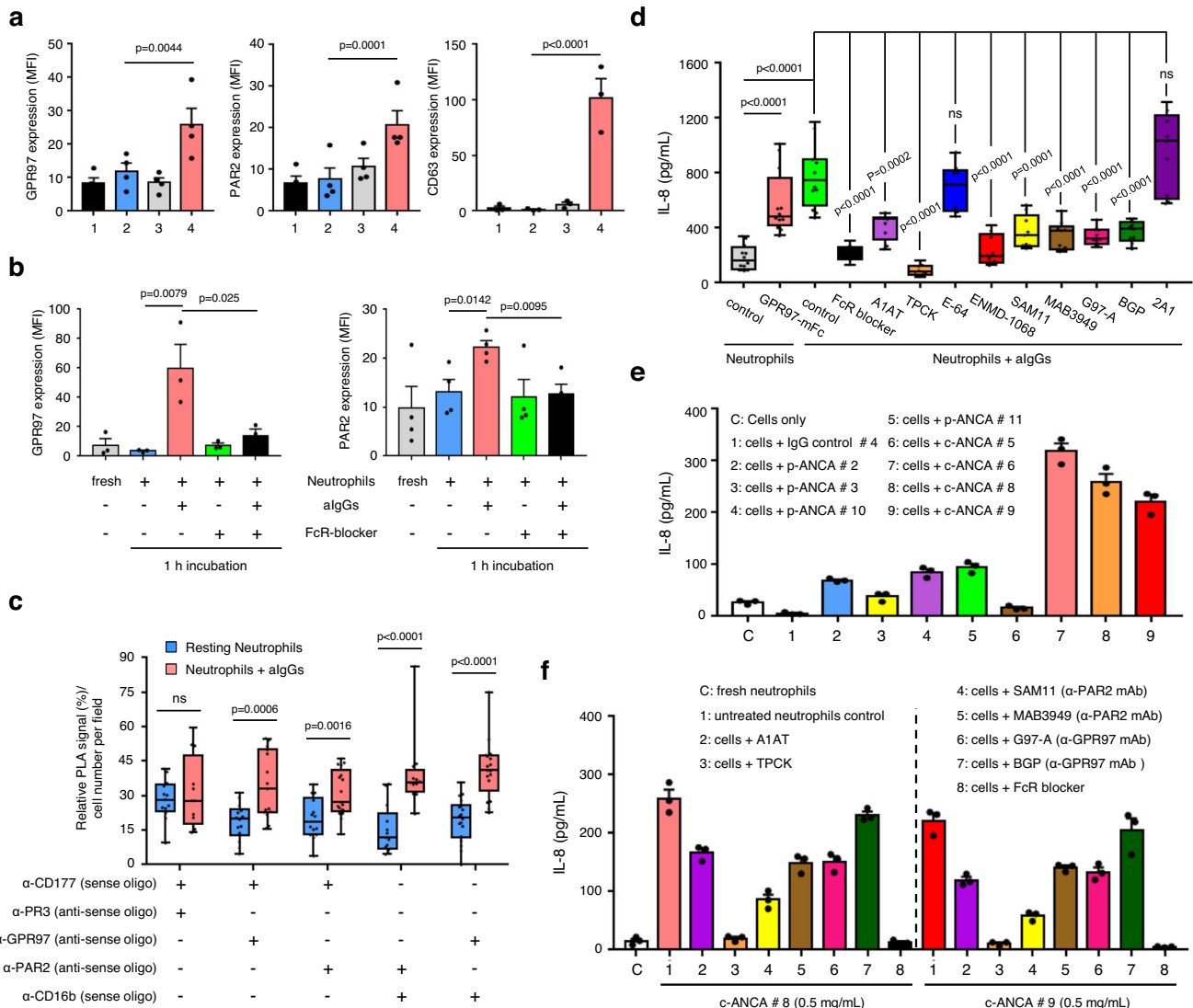

**Fig. 6 | Up-regulated GPR97 and PAR2 expression in activated neutrophils induces GPR97-PAR2 activation. a, b** Flow cytometry analyses of GPR97 and PAR2 expression in resting neutrophils treated without or with the azurophil granule degranulation stimulants (*n* = 4, except CD63 *n* = 3) **a** and aIgGs (GPR97 *n* = 3, PAR2 *n* = 4) **b** as indicated. **a** Samples include: 1, fresh neutrophils; 2, untreated neutrophils; 3, neutrophils treated with fMLF (1 μM) for 15 min; 4, neutrophils treated with cytochalasin B (cytoB, 5 μM) for 5 min, followed by fMLF (1 μM) for 10 min. CD63 expression was used as a degranulation marker of azurophil granule. **c** Relative PLA signals of indicated receptor pairs in resting (*n* = 3) and aIgG-treated

(*n* = 3) neutrophils. **d** ELISA analyses of IL-8 secreted by neutrophils incubated for 3 h at 37 °C as indicated (*n* = 3). Data in **a**–**d** are presented as means ± SEM. *P* value was determined by two-way ANOVA in **a**, **b**, **d** and by one-sided unpaired student's t-test in **c**. ns non-significant. **e, f** ELISA analyses of IL-8 secreted by neutrophils that were incubated with purified IgGs (0.5 mg/mL) from different PR3-ANCA and MPO-ANCA patients for 3 h at 37 °C as indicated. Data are presented as means ± SEM of one representative experiment done in triplicate. ENMD-1068: PAR2 antagonist; SAM11 and MAB3949: anti-PAR2 mAbs; G97-A and BGP: anti-GPR97 mAbs; 2A1: anti-EMR2 mAb. Source data are provided in the Source Data file.

## GPR97-PAR2 activation promotes neutrophil-mediated antimicrobial activity and endothelial cell activation and dysfunction

To investigate the potential innate immune function of GPR97-PAR2 activation, the effects of GPR97$^E$-mFc on the bacterial uptake and killing abilities of neutrophils were examined. As shown, the engulfment of several bacteria species including *E. coli, S. typhimurium*, and *S. aureus* was enhanced in neutrophils incubated with GPR97$^E$-mFc, but not the mFc control (Fig. 7a). Similarly, the bacterial killing assay showed that GPR97$^E$-mFc treatment significantly facilitates the elimination of *S. typhimurium* and *S. aureus* by neutrophils (Fig. 7b). We compared neutrophil activation phenotypes after application of PAR2-specific agonists and found that GPR97-mediated PAR2 activation is more effective overall, suggesting that it is a predominant PAR2 activation mechanism in neutrophils (Supplementary Fig. 9).

Uncontrolled and excess activated neutrophils are associated with adverse tissue damage seen in various inflammatory disorders[36]. To delineate the effect of GPR97-PAR2 activation on the vasculature, the activation and dysfunction of endothelial cells were investigated using the HUVEC-neutrophil co-culture system. HUVECs were significantly activated when co-cultured with neutrophils in the presence of GPR97$^E$-mFc (Fig. 7c). Moreover, GPR97$^E$-mFc treatment promoted endothelial permeability and dysfunction (Fig. 7d). Similar increased endothelial permeability was identified in HUVEC-neutrophil co-culture in the presence of aIgGs. FcR blocker, PAR2 antagonists, or functional blocking anti-PAR2 and anti-GPR97 mAbs attenuated this effect (Fig. 7e). We conclude that mPR3-mediated GPR97-PAR2 activation in neutrophils plays a role in anti-microbial responses and endothelial cell activation and dysfunction.

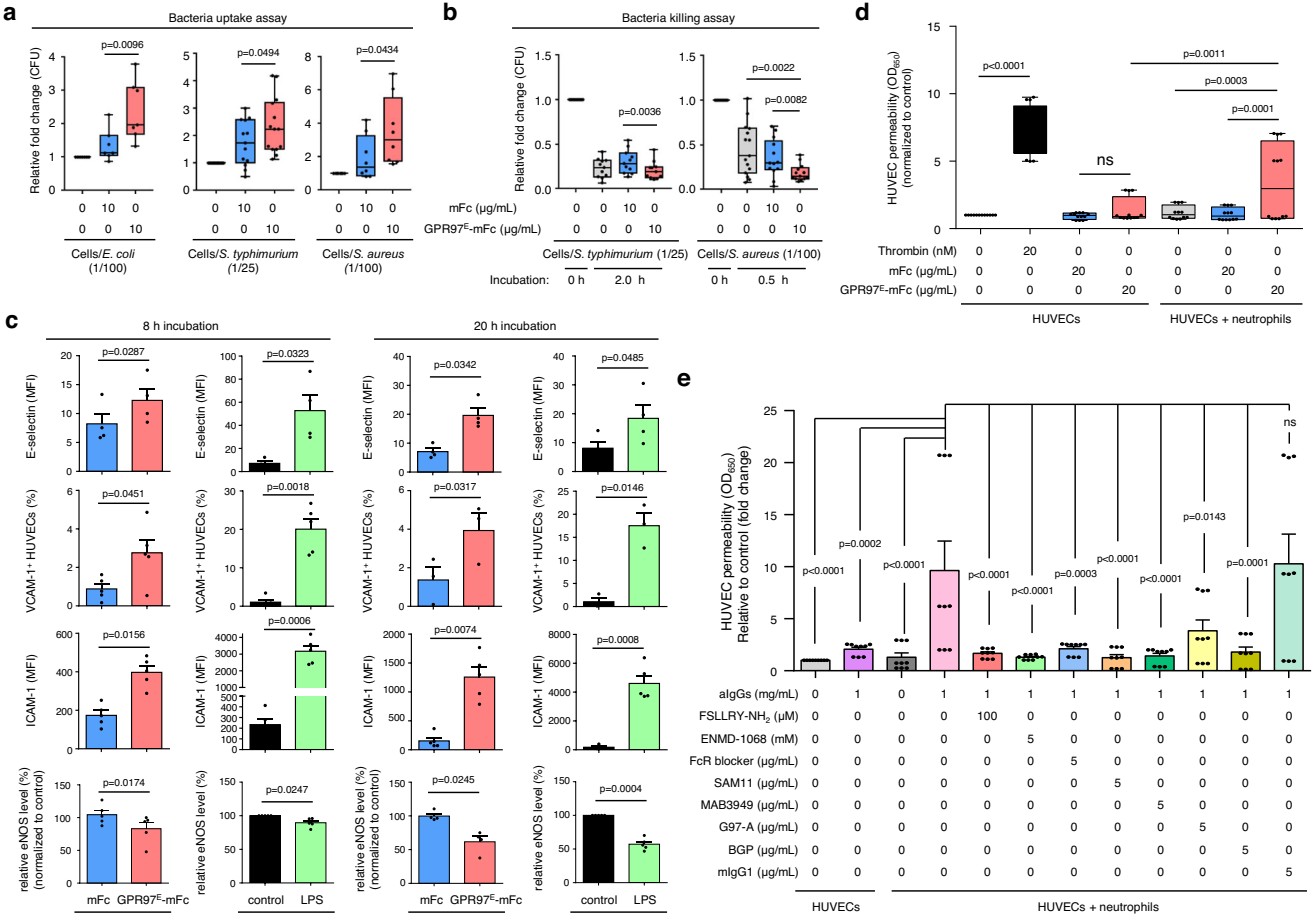

**Fig. 7 | GPR97-mediated PAR2 activation enhances neutrophil-mediated bacterial phagocytosis and killing as well as endothelial cell activation and dysfunction. a**, **b** Phagocytosis **a** and killing **b** of live bacteria (*E. coli* $n = 10$, *S. typhimurium* $n = 13$ for phagocytosis assay $n = 11$ for killing assay, and *S. aureus* $n = 8$ for phagocytosis assay $n = 11$ for killing assay) by neutrophils incubated without or with GPR97[E]-mFc. mFc was included as a negative control. Data are means ± SEM. *P* value was determined by one-way ANOVA. **c** The expressional analyses of cell activation markers (E-selectin, ICAM-1, VCAM-1 and eNOS) of HUVECs co-cultured with neutrophils in the absence or presence of GPR97[E]-mFc. mFc was included as a negative control. HUVECs treated without or with LPS were used as a negative and a positive control, respectively. $n = 3$ independent experiments. Data are means ± SEM and p value was determined by two-sided unpaired student's t-test.

**d**, **e** Endothelial cell permeability assays of HUVECs co-cultured with neutrophils. **d** Assays were done in HUVEC-neutrophil co-culture in the absence or presence of GPR97[E]-mFc for 20 h at 37 °C. mFc was included as a negative control. HUVECs alone treated without or with mFc were negative control groups, while those treated with thrombin were the positive control. $n = 4$ independent experiments. **e** The HUVEC-neutrophil co-culture was treated without or with aIgGs in the absence or presence of protease inhibitors/PAR2 antagonists/blocking Abs as indicated. HUVECs alone treated without or with aIgGs were included as controls. $n = 3$ independent experiments. Data are means ± SEM and *p* value was determined by two-way ANOVA. ns, non-significant. Source data are provided in the Source Data file.

## Discussion

In this study, GPR97 was identified as the binding partner and allosteric activator of mPR3 which in turn activated PAR2 on human neutrophils (Figs. 2 and 3). The mPR3-mediated GPR97-PAR2 activation adds a significant component to the repertoire of neutrophil activation mechanisms (Figs. 5–7).

PAR2 activation is normally induced by proteolytic enzymes including soluble proteases such as trypsin and microbial proteases as well as single-transmembrane or glycosylphosphatidylinositol-linked membrane sheddases such as matriptase and testisin[37,38]. The active coagulation protein complex containing tissue factor (TF)/factor VIIa (FVIIa)/factor Xa (FXa) could also act as a potent PAR2 activator[39]. In this case, PAR2 is transactivated indirectly by the TF/FVIIa/FXa complex via its proteolytic activation of the matriptase zymogen[40]. By contrast, PAR1-PAR2 transactivation was achieved via interaction of PAR2 with the thrombin-exposed tethered ligand of PAR1[14]. Interestingly, the GPR97-PAR2 activation process also involves a unique protein complex, here containing a serine

protease, two glycosylphosphatidylinositol-linked receptors, and an aGPCR (Fig. 5h).

PAR2 is expressed ubiquitously and its activation has been shown to play multiple immune-regulatory roles including neutrophil activation and inflammatory cytokine production[41–43]. Critically, neutrophil activation via the TF/FVIIa/PAR2 axis resulted in trophoblast injury and fetal death in an animal model of autoimmune antiphospholipid syndrome[11]. Our results herein show that neutrophil PAR2 activation, hereby GPR97/mPR3, plays a role in endothelial cell activation/dysfunction as well as bacterial uptake and less effectively in bacterial killing (Fig. 7). Our findings of the up-regulation of GPR97 and PAR2 expression in activated neutrophils suggested that the GPR97-PAR2 activation mode is likely a common neutrophil activation mechanism associated with inflammatory diseases (Figs. 1 and 6).

The absolute requirement of the clustered PR3/CD177/GPR97/PAR2/CD16b complex for efficient GPR97-PAR2 activation is surprising (Fig. 5), but it is not without precedent. In fact, it is highly analogous to the selective involvement of the glycosylphosphatidylinositol-

anchored RECK and GPR124/ADGRA2 receptor complex in mediating the Frizzled 4 (FZD4)/Low-density lipoprotein receptor-related protein 5 (LRP5)-dependent Wnt7a-specific bioactivity for CNS angiogenesis and blood-brain barrier integrity[44,45]. Coincidentally, the two aGPCR-GPCR protein complexes all consist of one soluble protein (PR3/Wnt7) and 4 receptor molecules including one glycosylphosphatidylinositol-linked receptor (CD177/RECK), one aGPCR (GPR97/GPR124), and one GPCR (PAR2/FZD4).

GPCR transactivation is an efficient means to expand the signaling outputs and usually involves the activation of another receptor type such as receptor tyrosine kinases (RTKs) by GPCRs[46]. Nevertheless, recent studies have shown that transactivation of GPCRs can also be induced by RTKs and other GPCRs[47,48]. GPCR transactivation of RTKs typically involves the activation of extracellular metalloproteases, which cleave and release membrane-bound protein ligands of RTKs[49]. In line with this, it is intriguing to note the involvement of GPR97-augmented mPR3 enzymatic activity in the PAR2 activation process (Fig. 3). Interestingly, approximate analogies were also noted in the Wnt7a-RECK-GPR124 complex in which RECK acted as a competitive inhibitor of Wnt7a signaling by specifically ligating and segregating Wnt7a from FZD4. GPR124 engagement efficiently realigned the RECK-Wnt7a complex to be available for FZD4 recognition hence permitting signaling[45]. With these in mind, we suggest that the aGPCR-GPCR activation mechanism symbolizes a previously unappreciated GPCR activation paradigm beyond the classical ligand-induced receptor activation.

GPR97 belongs to the ADGRG subgroup of the aGPCR family which contains some of the smaller-sized members including GPR56 and GPR114[50]. Although the GAIN domain has been identified as an evolutionarily conserved structural fold sufficient for GPS auto-proteolysis, the minimal domain structure needed for the unusual auto-proteolytic reaction remained to be fully elucidated[27]. Our analyses of GPR97-ECR showed that a well-conserved subdomain B plus a small one α-helix-containing subdomain A are competent for efficient GPS auto-proteolysis (Fig. 4). Recently, the cryo-electron microscopy structures of glucocorticoid-bound GPR97-G$_o$ complexes have revealed the first 3D structure of the seven-transmembrane region of an aGPCR[51]. With the structural resolution of GPR97-ECR reported here, it is possible now to build a complete structural feature of a full-length GPR97.

PR3 is the most abundant protease of azurophilic granules with versatile intracellular and extracellular functions[4]. Specific binding of mPR3 by PR3-ANCA which ligated and activated FcγRs was considered as the major disease mechanism of neutrophil activation in GPA[35]. Despite these, little is known of the physiological function of mPR3. Our findings of the tunable enzymatic activity of mPR3 by GPR97 and the subsequent PAR2 activation hence uncover a role for mPR3 in immune regulation (Figs. 3–7). Paradoxically, the outcomes of PR3-mediated PAR2 cleavage seemed to depend critically on the cell types studied. Indeed, PR3 digestion was shown to disarm PAR2 for subsequent trypsin-mediated activation in kidney epithelial cells[52]. On the other hand, PAR2 was efficiently activated by PR3 in immature dendritic cells[41] and by the IL-32γ/PR3 complex in THP-1 cells[53]. In the future, it could be a valuable advance to reveal the mechanistic detail of GPR97-PAR2 activation and to survey other potential protein substrates of the GPR97-augmented mPR3 activity.

In view of the unique human neutrophil-specific feature of GPR97-mPR3 interaction (Fig. 2, and Supplementary Fig. 1c), it was surprising to find interspecies differences in several members of the PR3/CD177/GPR97/PAR2/CD16b complex. Human PR3 differed from gibbon and murine PR3s by containing a distinctive hydrophobic patch which mediated its interaction with CD177[54]. Unlike the neutrophil-restricted expression of GPR97 in human, Adgrg3/Gpr97 was involved in B-lymphocyte fate decision[55], obesity-associated macrophage inflammation[56], and lymphatic endothelial cell migration in mice[57].

Finally, as CD16b is the neutrophil-specific glycosylpho-sphatidylinositol-anchored FcγRIII expressed only in human and not in mice[58], it suggests that GPR97-PAR2 activation is a human neutrophil-restricted phenomenon thereby highlighting the requirement for caution when extrapolating mouse data. Importantly, the strict regulation of GPR97 and PAR2 expression in neutrophils indicates that the GPR97-PAR2 activation reaction is induced only by unique inflammatory triggers, such as FcR-mediated signalling, that up-regulate surface GPR97 and PAR2 expression to a significant level. We believe the reason for having the PR3/CD177/GPR97/PAR2/CD16b complex, consequently a GPR97-PAR2 activation mechanism, is to provide an extra layer of control that acts on primed/activated neutrophils. Once induced, it allows for full neutrophil activation. Our results suggest that this is important for an effective anti-microbial response and immune effector function. Moreover, as proteolytic enzymes and GPCRs belong to two major families of molecular drug targets, the PR3/CD177/GPR97/PAR2/CD16b receptor complex represents a potential multi-target complex for the development of therapeutics to modulate human neutrophil-mediated inflammatory diseases[59,60].

## Methods

### Reagents and antibodies

All chemicals and reagents were purchased from Sigma (St. Louis, MO, USA) and Invitrogen (CA, USA) unless specified otherwise. E-64 (#78434) was from Thermo Fisher Scientific. Human proteinase 3 (PR3) (#16-14-161820) was purchased from Athens Research & Technology, Inc. (Athens, Georgia, USA). Human Fc receptor blocker (Human BD Fc Block™, #564220) was from BD Biosciences (New Jersey, USA). Antibodies used in this study are listed in Supplementary Table 1.

### Cell culture

All cell culture media and supplements, including fetal calf serum (FCS), L-glutamine, penicillin and streptomycin were purchased from Invitrogen. Cell lines used in this study (listed in Supplementary Table 2) were purchased from the American Type Culture Collection (Manassas, VA, USA) or Bioresource Collection and Research Center (Hsinchu, Taiwan) and cultured in conditions as suggested.

### Construction of expression vectors and cell transfection

Mammalian expression constructs encoding the mFc only as well as various GPR97-, GPR56-, and EMR2-mFc fusion proteins were generated in the pcDNA3.1-mFc vector using standard molecular biology technologies as described previously[19]. Gene-specific oligonucleotide primers used to construct the mFc-fusion protein expression vectors are listed in Supplementary Table 3. For the construction of expression vectors encoding GPR97-TM7-EGFP and GPR97-myc, the full-length human GPR97 cDNA was cloned into the pEGFP-N1 (Clontech) and pcDNA3.1/myc-His A (Invitrogen) vectors, respectively. Expression vectors encoding the PAR2, EPCR, Mac-1 (CD11b/CD18), CD16b, and CD55 receptors were purchased from Sino Biological Inc. (Beijing, China). The CD177 and FPR1 expression constructs were gifts from Dr. Ralph Kettritz of Charité of Universitätsmedizin Berlin, Germany and Dr. TL Hwang of Chang Gung University, Taiwan, respectively. The EMR2 expression construct was described previously[61]. For transient cell transfection, HEK-293T cells were transfected with purified plasmid DNAs using Lipofectamine™ or Lipofectamine™ 2000 (Invitrogen) as described elsewhere.

### Generation and purification of the mFc-fusion proteins

All recombinant mFc-fusion proteins were purified from conditioned media of transiently transfected HEK-293T cells using the Protein A-Sepharose affinity chromatography as described previously[19]. Briefly, HEK-293T cells were transfected with the expression constructs of interest using the calcium phosphate-based transfection method. Following transfection, cells were cultured in serum-free

OPTI-MEM medium for 4–5 days and conditioned media was collected, centrifuged, and passed through the Protein A-Sepharose affinity column (nProtein A Sepharose™ 4 Fast Flow, GE Healthcare), followed by extensive washes with washing buffer (50 mM Tris-HCl pH 7.4, 10 mM CaCl$_2$, 150 mM NaCl). The mFc-fusion protein was eluted and subjected to dialysis in the Slide-A-Lyzer® Dialysis Cassettes (Thermo Scientific) in washing buffer for 24 h at 4 °C, filtered, and stored at −80 °C until use.

## Purification of blood neutrophils

Human neutrophils were isolated from fresh venous blood donated by healthy volunteers and diseased patients who have all given the signed informed consent and receive no participant compensation. All experimental procedures were approved by the Chang Gung Memorial Hospital Ethics Committee (CGMH IRB No: 201701852B0, 201901358B0, 201901293B0, and 202002255B0) and performed according to the guidelines set by the Committee. In brief, blood samples were drawn into the collection tubes coated with sodium heparin and neutrophil isolation was performed using Polymorphprep™ density gradient centrifugation (Axis-Shield, Oslo, Norway) as described previously[18]. For the isolation of murine neutrophils, blood was collected by cardiac puncture and separated by Ficoll (GE Healthcare) gradient separation. Following the lysis of the remnant erythrocytes, isolated neutrophils were checked for purity by flow cytometry with specific surface markers and resuspended in RPMI medium containing 10% FCS for all following experiments unless otherwise denoted. The isolation procedure routinely produced a > 95% pure and viable neutrophil population. When indicated, isolated human neutrophils were further subjected to the magnetic cell sorting (MACS) separation of CD177$^+$ and CD177$^-$ sub-populations using PE-conjugated anti-CD177 Ab coupled to the MACS MicroBeads and Separator (Miltenyi Biotec, Bergisch Gladbach, Germany) according to the manufacturer's recommendations.

## Immunofluorescence assay and immunohistochemistry

Freshly isolated human neutrophils (1 × 10$^6$ cells/mL) were treated as described and then fixed with 4% paraformaldehyde/PBS for 20 min at 4 °C. Cells were then incubated with the blocking buffer (1% BSA, 5% normal goat serum (NGS) in PBS) for 1 h at 4 °C, then with indicated primary antibody (5 μg/mL) diluted in blocking buffer for 1 h at 4 °C. Following extensive washes in cold PBS, cells were incubated for 1 h at 4 °C with fluorescence-labeled secondary antibody diluted in blocking buffer at a pre-determined optimal concentration. Cells were washed thoroughly before being spun onto slides. Fluorescence images were taken by Fluoview FV10i (Olympus) or the LSM780 confocal microscope system (ZEISS). Immunohistochemistry was carried out on formalin-fixed, paraffin-embedded tissue sections (4 μm) using G97-A anti-GPR97 mAb (5 μg/mL) as described previously[18]. All tissue staining procedures were performed in an automated immunostainer (BOND-MAX™, Leica Biosystem). Immunoreactivity was assessed independently by two expert pathologists. Neutrophils in capillaries were found to display strong GPR97 expression constantly. Therefore, the GPR97 expression level of tissue infiltrating neutrophils was scored as follows: 1 + , if GPR97 expression was far weaker than that of intravascular neutrophils; 2 + , if GPR97 expression was weaker than that of intravascular neutrophils; 3 + , if the GPR97 expression intensity was equal to that of intravascular neutrophils; 4 + , if GRP97 expression was stronger than that of intravascular neutrophils.

## Flow cytometry analysis and the FACS-based ligand-binding assay

All procedures were performed at 4 °C. Cells were blocked in ice-cold blocking buffer (1% BSA/5% NGS in PBS) for 1 h. For the standard flow cytometry analysis, cells were incubated sequentially with the primary Ab of interest and appropriate fluorescence-labeled secondary Ab diluted in blocking buffer at pre-determined optimal concentrations for 1 h. For the analysis of GPR97 expression in blood neutrophils of normal controls and diseased patients, blood leukocytes were obtained following RBC lysis with ACK buffer and washed twice with ice-cold PBS. Cells (1 × 10$^6$ cells/reaction) were stained with a mixture of aqua fluorescent reactive dye (MAN0006891, Life Technologies), anti-CD16b-FITC (130-126-529, Miltenyi Biotec), anti-CD66b-PerCP-Cy5.5 (305108, Biolegend), and G97-A conjugated with APC (Ab201807, Abcam) for 20 min. Fluorescent minus one controls include GPR97 and aqua fluorescent reactive dye only control were included for background determination. The fluorescence-labeled cells were analyzed by BD FACSCanto II flow cytometry system. For the cellular ligand-binding assay, cells were incubated with the mFc-fusion protein probe (10 μg/mL) in blocking buffer for 2 h. Where indicated, cells were pretreated or treated simultaneously with various reagents in the presence of the probes. Fluorescence-conjugated goat anti-mouse IgG (2 μg/mL) was used as the 2$^{nd}$ Ab. Following extensive washes between each incubation step, cells were washed lastly in cold PBS and subjected to analysis by FACSCalibur flow cytometer (BD Biosciences). Positive ligand-binding was determined as the shift of fluorescence signals. The mFc protein was always included as a negative control.

## Phenotypic analyses of neutrophils

Freshly isolated human neutrophils (1 × 10$^6$ cells/mL) were suspended in RPMI complete medium and incubated with soluble GPR97$^E$-mFc, control mFc protein (10 μg/mL), or f-MLF (10$^{-6}$ M) for 3 h at 37 °C. For the expressional analysis of CD markers, cells were fixed by 4% paraformaldehyde/PBS for 20 min at 4 °C and stained with fluorescence-conjugated CD markers as indicated using the standard flow cytometry protocol. For the analysis of neutrophil morphological changes, live cells were either collected at 1 h after incubation and photographed under the inverted microscopy, or analyzed directly by FACSCalibur flow cytometer and shown as FSC/SSC. Alternatively, cells were fixed and subjected to the Wright-Giemsa stain, or permeabilized and blocked in the blocking buffer (1% BSA, 5% NGS, 0.1% saponin in PBS) for 30 min, then incubated with Phalloidin-TRITC (10 μg/mL) and analyzed by flow cytometer as well as Fluoview FV10i (Olympus). For IL-8 production, culture supernatant was collected at 3 h incubation and subjected to the ELISA analysis.

## Analyses of reactive oxygen species production and myeloperoxidase activity

Freshly isolated human neutrophils (2 × 10$^6$ cells/mL) were resuspended in PBS supplemented with 0.2% BSA and 5 mM glucose, and incubated with 2 μM dihydrorhodamine-123 (DHR123; Molecular Probes #D23806) for 30 min at room temperature (RT). Cells were then incubated respectively with GPR97$^E$-mFc or control mFc-fusion protein (10 μg/mL) for 30 min at 37 °C. Cells were then stimulated with or without fMLF (10$^{-6}$ M) for 15 min before being placed on ice to stop the reaction. The accumulation of ROS represented by fluorescent oxidized DHR123 was immediately measured by flow cytometry as described previously[18].

The MPO enzyme activity was analyzed using the colorimetric MPO activity assay kit (BioVision) performed exactly as suggested by the manufacturer. Briefly, freshly isolated human neutrophils (2.5 × 10$^6$ cells/mL) were resuspended in PBS supplemented with 0.2% BSA and 5 mM glucose. Cells were then incubated with GPR97$^E$-mFc or control mFc fusion protein (10 μg/mL) for 30 min at 37 °C. When necessary, cells were first incubated with GPR97$^E$-mFc, followed by stimulation with or without fMLF (10$^{-6}$ M) for 15 min. Cells were lysed with the MPO assay buffer and then centrifuged at 10,000 $g$ for 10 min at 4 °C. Lysate supernatant (10 μL/sample) was added to 40 μL of MPO assay buffer in a 96-well plate. The reaction was initiated by the addition of 10 μL MPO substrate solution to each well and incubated for 60 min at RT. Finally,

2 μL of stop solution was added, followed immediately with 50 μL of Ellman's reagent (5,5′-dithiobis-(2-nitrobenzoic acid), DTNB) for the colorimetric analysis of the MPO activity at 412 nm.

## Measurement of IL-8 by sandwich ELISA

Freshly isolated human neutrophils ($1 \times 10^6$ cells/mL) were cultured in RPMI/10% FBS in the presence of fMLF ($10^{-6}$ M), mFc or GPR97[E]-mFc (10 μg/mL) for 3 h at 37 °C. When indicated, culture plates were coated with the mFc fusion proteins at 4 °C overnight before incubation with neutrophils. When necessary, neutrophils were pretreated with various reagents/inhibitors such as α1-antitrypsin (5 mg/mL), aprotinin (50 μg/mL), E-64 (50 μg/mL) and N-p-Tosyl-L-phenylalanine chloromethyl ketone (TPCK, 10 μM) for 1 h at 37 °C or incubated with elafin (4 μM) and SLPI (4 μM) simultaneously. Cell-free supernatants were collected to determine the concentration of IL-8 by using the ELISA kit (DuoSet® ELISA, R&D Systems, DY208) according to the protocols suggested by the manufacturer.

## Structural analysis of GPR97-ECR

For crystallisation experiments, human GPR97-ECR (UniProt Q86Y34, residues 1 to 264) was cloned into the *Eco*RI-*Kpn*I cloning sites of vectors from the pHLsec family with its native secretion signal sequence and a C-terminal His[6] tag. The cDNA encoding GPR97 possesses a SNP at residue 447 (M→V). GPR97-ECR was expressed in a secreted form in GlnTI-deficient HEK-293S cells using previously described protocols[62]. Briefly, plasmid DNAs were transfected with PEI in a 1:2 ratio into 3 L of 90–100% confluent HEK-293S cells. After 10 days, the cell culture medium containing the secreted GPR97-ECR was clarified by centrifugation and filtration prior to diafiltration into PBS, 20 mM Tris pH 7.5 and 150 mM NaCl. GPR97-ECR was then purified by immobilized metal affinity chromatography (HiTrap™ HP, GE Healthcare) and size-exclusion chromatography (Superdex 200 16/600, GE Healthcare) in 20 mM Tris pH 7.5 and 300 mM NaCl. The protein purity was assessed by sodium dodecyl sulphate-polyacrylamide gel electrophoresis (SDS-PAGE) and concentrated to 40 mg/mL using 10,000 MWCO concentrators (Amicon Ultra Centrifugal Filters).

GPR97-ECR crystals grew in 100 nL + 100 nL sitting nanodrops by the vapour diffusion method at 18 °C in 1.8 M tri-ammonium citrate pH 7. The crystals were harvested and cryo-protected in 3 M tri-ammonium citrate pH 7. A native dataset was collected at a wavelength of 0.9763 Å at the European Synchrotron Radiation Facility beamline ID30B. 180° of data were collected with an exposure of 0.02 s per 0.1° rotation on a PILATUS3 6 M (Dectris). The data was processed using DIALS[63] and AIMLESS[64]. A S-SAD dataset was also collected at a wavelength of 2.75 Å on the long-wavelength beamline I23[65] at Diamond Light Source, UK. 360° of data were collected with an exposure of 0.1 s per 0.1° rotation on a PILATUS 12 M detector (Dectris). The data showed sign of anisotropy and was therefore processed using AUTOPROC[66] and STARANISO (Tickle, I.J. et al. (2018) STARANISO. Cambridge, United Kingdom: Global Phasing Ltd.) (Supplementary Table 4).

The flexible loops of the GAIN domains of Lphn1, BAI3 and GPR56 (PDB accession numbers 4DLQ, 4DLO, 5KVM respectively) were removed and an ensemble of those trimmed GAIN domains was used as a search model for molecular replacement using the native dataset and PHASER[67]. The solution consisted of two molecules in the asymmetric unit and displayed additional density on top of the GAIN domain, accounting for the small N-terminal domain. To help build this N-terminal domain, molecular replacement using the S-SAD dataset and PHASER[67] was carried out with the partial model consisting of the GPR97 GAIN domain. The resulting solution was subjected to one round of refinement in autoBUSTER (Bricogne G., et al (2017) BUSTER version 2.10.3. Cambridge, United Kingdom: Global Phasing Ltd.)[68]. Electron density maps were improved in PARROT[69] revealing the missing N-terminal domain. Anomalous difference maps calculated with ANODE[70] were used to locate sulphur atoms and help manual building in COOT[71]. All-atom refinement with autoBUSTER (Bricogne G., et al (2017) BUSTER version 2.10.3. Cambridge, United Kingdom: Global Phasing Ltd.)[68] resulted in the final model containing residues 28–260.

## Western and far-western immunoblotting analysis

Cells were lysed in RIPA lysis buffer (20 mM Tris-HCl pH 7.4, 5 mM MgCl$_2$, 100 mM NaCl, 0.5% Nonidet P-40, and 1× Complete Protease Inhibitors) supplemented with 1 mM sodium orthovanadate, 1 mM AEBSF, and 5 mM Levamisole. Lysate proteins were quantified using the Bicinchoninic acid (BCA) protein assay kit (Pierce). Protein samples were separated in 10% SDS or native PAGE gels and transferred to PVDF membranes. Membranes were then soaked in blocking buffer (PBS containing 5% nonfat skimmed milk and 0.1% Tween 20) for 1 h at RT. Next, the membranes were incubated with indicated primary Ab or GPR97[E]-mFc (5 μg/mL) diluted in blocking buffer overnight at 4 °C. EMR2[E]-mFc or mFc (5 μg/mL) protein was used as negative controls. Following extensive washes in washing buffer (0.1% Tween 20 in PBS), membranes were incubated with horseradish peroxidase (HRP)-conjugated goat anti-mouse Fc Ab (1:2000) in blocking buffer. Finally, the specific binding of Ab and mFc-fusion proteins on membrane blots was revealed with chemiluminescent HRP Substrate (Millipore WBKLS0500).

## Protein extraction of the cytoplasmic and membrane fractions

All protein isolation procedures are performed at 4 °C. Freshly isolated neutrophils ($2 \times 10^7$ cells) were washed three times with cold PBS, resuspended in RIPA buffer and passed through a 30-gauge needle 20 times. Cell lysates were then centrifuged at 700 *g* for 3 min at 4 °C, and the supernatant was collected and subjected to ultracentrifugation at 10,000 *g* for 40 min at 4 °C. The resulting supernatant was collected as the cytoplasmic fraction and the pellet was the membrane fraction. All protein samples were stored at −80 °C until use.

## FACS-based PR3-binding assay

HEK-293T cells were transfected with various expression constructs as indicated. HEK-293T transfectants were collected, washed, and incubated in blocking buffer (1% BSA, 5% NGS/PBS) for 1 h at 4 °C before incubating with commercially available purified PR3 (5 μg/mL) diluted in blocking buffer for 1 h at 4 °C. Cells were subjected to extensive washes in cold PBS, and then incubated with the anti-PR3 mAb (5 μg/mL)(clone MCPR3-2, Thermo Fisher Scientific MA5-11945) or PeliCluster ANCA (5 μg/mL)(clone CLB-12.8, Sanquin M1574) for 1 h at 4 °C. Cells were extensively washed and then incubated with fluorescence-labeled secondary antibody for 1 h at 4 °C. Finally, cells were washed three times in cold PBS and subjected to analysis by FACSCalibur flow cytometer (BD Biosciences). When indicated, HEK-293T cells were first transfected with the GPR97-TM7-EGFP construct. Transfected cells were then subjected to FACS sorting by FACSAria™ II (BD Biosciences) to select the top 30% highest GPR97-expressing (GPR97[high]) and low GPR97-expressing (GPR97[low]) cells based on the GFP expression levels. The sorted GPR97[high] and GPR97[low] HEK-293T cells were used to examine the PR3-binding ability as described above. Similar EMR2[high] and EMR2[low] HEK-293T transfectants were included as negative controls.

For the analysis of GPR97-mPR3 binding in transfected HEK-293T cells, cells were transfected as described. Half of the transfected cells were checked by the flow cytometry analysis using receptor-specific Abs for the expression of the specific receptors. The other half of the transfected cells were incubated first with purified PR3 (5 μg/mL) diluted in blocking buffer for 1 h at 4 °C, washed, and followed by incubation with mFc (10 μg/mL) or GPR97[E]-mFc (10 μg/mL) diluted in blocking buffer for 2 h at

4 °C. Cells were washed and reacted with fluorescence-labeled goat anti-mouse IgG for 1 h at 4 °C. Finally, cells were washed extensively in cold PBS and subjected to analysis by FACSCalibur flow cytometer (BD Biosciences).

### ELISA-like PR3-binding assay

Saturated mFc-fusion proteins (10 µg/mL), including CD177-mFc, GPR56[E]-mFc, GPR97[E]-mFc, and mFc were separately coated onto 96-well plates for 16 h at 4 °C. Coated plates were washed with the wash buffer (0.05% Tween 20 in PBS) three times, and blocked with the blocking buffer (5% BSA in PBS) for 1 h at 4 °C. Purified PR3 protein (1, 1.5, 2, and 3 µg/mL in blocking buffer) was added for 2 h at 4 °C. Wells were then washed extensively and incubated with the MCPR3-2 anti-PR3 mAb (0.4 µg/mL in blocking buffer) for 2 h at 4 °C. Wells were washed extensively and incubated with goat anti-mouse κ chain HRP (1 µg/mL) for 1 h at 4 °C. Following extensive washes, tetramethylbenzidine (TMB) substrate (100 µL/well) was added for 20 min at 4 °C before adding the stop solution (2 N $H_2SO_4$)(50 µL/well). Reaction signals were detected by the ELISA reader at OD450.

### The ex vivo membrane PR3 activity assay

The ex vivo neutrophil membrane PR3 activity was measured using the PR3-specific FRET substrate, Abz-VADnV-RDRQ-EDDNP (Cayman, MI, USA) exactly as described previously[26]. Briefly, freshly isolated neutrophils (1 × 10^6 cells/well) were washed twice with PBS. Cells were then suspended in 150 µL PBS and incubated with recombinant mFc-fusion proteins of interest in appropriate concentration for 10 min at 37 °C before adding the PR3-specific FRET substrate (20 µM). Fluorescence intensity was detected by the SpectraMax M2e ELISA reader (Molecular Devices, CA, USA) for at least 90 min during the reaction. When necessary, cells were treated with various protease inhibitors as indicated. The corresponding Vmax was analyzed by the Softmax Pro 5.3 software.

### The proximity-ligation assay

To detect the close interaction of CD177, PR3, GPR97, PAR2, and CD16b in human neutrophils, the Duolink in situ PLA (Sigma-Aldrich®, DUO92013) was employed according to the manufacturer's protocol. Purified human neutrophils were cytospun on Poly-D-Lysine-coated coverslips and blocked in blocking buffer (1% BSA/5% NGS in PBS) for 60 min at 4 °C. Cells were incubated with the receptor-specific Abs (15 µg/mL) pre-conjugated with the PLA-PLUS or -MINUS oligonucleotide probe (Sigma-Aldrich®, DUO92009 and DUO92010) for 60 min at RT. The enzymatic ligation of the PLA-PLUS and −MINUS probes were performed for 30 min at 37 °C, followed by circle amplification for 90 min at 37 °C and finally incubated with Hoechst 33342 (Invitrogen) for nuclear staining. Cells were observed by the LSM780 confocal microscope system (ZEISS) at 40× magnification.

### Bacteria uptake and killing assays

Freshly isolated neutrophils (1.25 × 10^6 cells/mL) were suspended in RPMI and incubated without or with GPR97[E]-mFc or control mFc fusion protein (10 µg/mL) respectively for 30 min at 37 °C. Neutrophils were then incubated with live *E. coli* (DH10B)(MOI 1:100), *S. typhimurium* (MOI 1:25), and *S. aureus* (MOI 1:100) in logarithmic-phase for 1 h at 37 °C. For the bacteria uptake assay, extracellular bacteria were removed by extensive washing in PBS, followed by incubation for 30 min in RPMI-1640 medium containing 50 µg/mL gentamicin. Afterwards, neutrophils were washed twice with PBS and lysed with 0.05% Triton X-100 in PBS. For the bacteria killing assay, neutrophils were collected and lysed with 0.05% Triton X-100 in PBS at desired time points after the bacteria uptake and the removal of extracellular bacteria by gentamicin. Serial dilutions of the neutrophil lysate were plated on LB plates and incubated overnight at

37 °C to determine the bacterial colony-forming units (CFU) the next day.

### Neutrophil-HUVEC co-culture

Human umbilical vein endothelial cells (HUVECs) were maintained in Medium 199 supplemented with 10% FBS, L-glutamine, penicillin, streptomycin, 25 U/mL heparin and 30 µg/mL endothelial cell growth supplement. HUVECs (6 × 10^5 cells/well) were plated on 1% gelatin-coated culture dishes (6 mm) until confluence. HUVECs were incubated with freshly isolated human neutrophils (6 × 10^6 cells/well) without or with indicated mFc-fusion proteins (10 µg/mL) for 8 or 20 h at 37 °C. HUVECs incubated with LPS (1 µg/mL) were used as a positive control. Cells were washed with ice-cold PBS twice and collected for analysis using flow cytometry by staining with CD62E-APC (E-selectin, BD Pharmingen), CD54-APC (ICAM-1, BD Pharmingen), CD106-PE (VCAM-1, BD Pharmingen) and eNOS-PE (BD Pharmingen). For the endothelial cell permeability assay, HUVECs (5 × 10^4 cells/well) were plated on 1% gelatin-coated Transwell plates (12 mm)(Costar, Corning) containing polycarbonate membranes (3 µm pore size) until confluence. Cells were incubated with freshly isolated human neutrophils (1 × 10^6 cells/well) without or with indicated mFc-fusion proteins (20 µg/mL) for 24 h. HUVECs incubated with thrombin (20 nM) for 30 min were used as a positive control. Alternatively, HUVECs were incubated with human aIgGs (1 mg/mL) without or with PAR2 antagonists (FSLLRY-NH₂ and ENMD-1068)(100 µM), PAR2 antibodies (SAM11 and MAB3949)(5 µg/mL), GPR97 antibodies (G97-A and BGP)(5 µg/mL) and mouse IgG1 as indicated for 8 h. The lower chambers were replenished with fresh M199 growth medium while the upper chambers were filled with fresh medium containing 4% BSA and Evans blue (0.67 mg/mL). Cells were incubated for 30 min at 37 °C and the optical density of the medium from lower chambers was measured at 650 nm in a spectrophotometer.

### Statistics and reproducibility

No statistical method was used to predetermine sample size and no data were excluded from the analyses. All results were analyzed using GraphPad Prism (version 6.0 and 7.0; GraphPad Software, San Diego, CA, USA) and expressed as means ± standard error of the mean (SEM) with the number of experimental replicates (n) provided. Differences between groups were determined by student's t-test, one-way, and two-way ANOVA as indicated. In all cases, a probability (p) value of <0.05 was accepted to reject the null hypothesis and considered significant.

### Reporting summary

Further information on research design is available in the Nature Research Reporting Summary linked to this article.

## Data availability

All data needed to evaluate the conclusions in the paper are present in the main manuscript, the Supplementary Information and the Source data. Source data are provided with this paper and comprise all relevant raw data from each figure in the main manuscript and in the Supplementary Information. The X-ray structure data of GPR97 extracellular domain generated in this study have been deposited in the database of Protein Data Bank in Europe (PDBe) under accession code PDB ID 7QU8 (Deposition ID: D_1292120102). Source data are provided in this paper.

## Code availability

No custom codes/softwares were generated in this study. The following publicly available softwares were used for structural analysis. DIALS 1.9.3 (https://dials.github.io/installation.html), Coot 0.9.6, Aimless 0.7.3, Phaser 2.8.3, Parrot 1.0.4, and CCP4 suite 7.1 (https://www.ccp4.ac.uk/download/#os=mac). ANODE 2013/1 is available through

the SHELX software suite at https://www.ccp4.ac.uk/download/#os=mac. autoPROC 1.0.5 and autoBUSTER 2.10.4 are from https://www.globalphasing.com/. STARANISO V3.347 is accessed at https://staraniso.globalphasing.org/. Structural figures and structural alignments were generated using PyMOL v.2.3.4. (https://pymol.org/2/). The AlphaFold model of GPR97 is available at https://alphafold.ebi.ac.uk/entry/Q86Y34. FlowJo v7.6.1 (https://www.flowjo.com/solutions/flowjo/downloads), ZEISS ZEN 2011 (https://www.zeiss.com/microscopy/int/products/microscope-software/zen.html#downloads), Olympus FluoView FV10i olympus fluoview fv10i (https://www.olympus-lifescience.com/en/support/downloads/), and Graphpad prism v6 and v7 (https://www.graphpad.com/scientific-software/prism/) were used for the analysis of cellular phenotypes.

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

## Acknowledgements

We thank Dr. Jörg Hamann and Dr. Gin-Wen Chang for their critical reading of the manuscript. The authors acknowledge the technical assistance from the Microscopy and Instrumentation Centers, Chang Gung University, Taoyuan, Taiwan. C.Z.G. was funded by the Wellcome Trust DPhil in Structural Biology programme. E.S. was funded by the Wellcome Trust (202827/Z/16/Z), the M.R.C. (MR/L018039/1), and the EMBO Young Investigator programme. H.H.L. was funded by Chang Gung Memorial Hospital (CMRPD1M0031, CMRPD1M0511, CMRPD1M0521, and CMRPD1M0321) and the Ministry of Science and Technology (MOST), Taiwan (MOST-107-2320-B-182-006 and MOST-110-2320-B-182-024).

## Author contributions

T.Y.C., C.Z.-G., K.Y.H., Y.C.C., Y.W.C., K.Y.I., Y.L.L., N.Y.C., H.Y.C., W.Y.T., C.Y.S., Y.M.W., Y.S.P., C.H.H., T.C.C., K.E.O., M.A., and S.R.H. performed experiments, M.S., S.G., C.Y.L., and A.S. provided advice and critical reagents, T.Y.C., C.Z.-G., E.S., and H.H.L. designed experiments and wrote the manuscript.

## Competing interests

The authors declare no competing interests.

## Additional information

Tai-Ying Chu ⓘ[1,14], Céline Zheng-Gérard ⓘ[2,14], Kuan-Yeh Huang[1], Yu-Chi Chang[1], Ying-Wen Chen[1], Kuan-Yu I[1], Yu-Ling Lo[1], Nien-Yi Chiang[1], Hsin-Yi Chen[1], Martin Stacey[3], Siamon Gordon[1,4], Wen-Yi Tseng[5], Chiao-Yin Sun[6,7], Yen-Mu Wu[8,9], Yi-Shin Pan[10], Chien-Hao Huang[10], Chun-Yen Lin[10], Tse-Ching Chen ⓘ[11], Kamel El Omari ⓘ[12], Marilina Antonelou[13], Scott R. Henderson[13], Alan Salama ⓘ[13], Elena Seiradake ⓘ[2,15] ✉ & Hsi-Hsien Lin ⓘ[1,5,11,15] ✉

[1]Department of Microbiology and Immunology, College of Medicine, Chang Gung University, Taoyuan, Taiwan. [2]Department of Biochemistry, University of Oxford, Oxford, UK. [3]Faculty of Biological Sciences, School of Molecular and Cellular Biology, University of Leeds, Leeds, UK. [4]Sir William Dunn School of Pathology, University of Oxford, Oxford, UK. [5]Division of Rheumatology, Allergy and Immunology, Chang Gung Memorial Hospital-Keelung, Keelung, Taiwan. [6]Department of Nephrology, Chang Gung Memorial Hospital-Keelung, Keelung, Taiwan. [7]Department of Medicine, College of Medicine, Chang Gung University, Taoyuan, Taiwan. [8]Graduate Institute of Clinical Medical Sciences, College of Medicine, Chang Gung University, Taoyuan, Taiwan. [9]Division of Infectious Diseases, Department of Internal Medicine, Chang Gung Memorial Hospital-Linkou, Taoyuan, Taiwan. [10]Department of Gastroenterology and Hepatology, Chang Gung Memorial Hospital-Linkou, Taoyuan, Taiwan. [11]Department of Anatomic Pathology, Chang Gung Memorial Hospital-Linkou, Taoyuan, Taiwan. [12]Diamond Light Source Limited, Harwell Science and Innovation Campus, Didcot, UK. [13]Department of Renal Medicine, Royal Free Campus, UCL, London, UK. [14]These authors contributed equally: Tai-Ying Chu, Céline Zheng-Gérard. [15]These authors jointly supervised this work: Elena Seiradake, Hsi-Hsien Lin. ✉e-mail: elena.seiradake@bioch.ox.ac.uk; hhlin@mail.cgu.edu.tw

