## [Peer Review File · Nature Communications]

GPR97 triggers inflammatory processes in human neutrophils via a macromolecular complex upstream of PAR2 activationREVIEWER COMMENTS

Reviewer #1 (Remarks to the Author):

This is a huge and well-developed study showing that a neutrophil surface receptor protein called GPR97 activates a protein complex on neutrophils, ultimately causing the activation of a different receptor, PAR2, and this in turn activates neutrophils. The report includes a crystal structure of the extracellular domain of GPR97, and a detailed molecular analysis of the interaction of this domain with the neutrophil cell surface protein complex. The writing is mostly clear, and the statistical analysis is good. The key thing missing from this report is the simple interpretation: what ultimately activates this mechanism, and why does this mechanism exist?

Major points

Page 6/ fig 2: GPR97 is expressed on the plasma membrane of neutrophils, and you show that the extracellular domain of GPR97 activates neutrophils, so you need to explain why two neutrophils touching each other won't activate each other.

If on the other hand you are arguing that GPR97 on a neutrophil activates the same neutrophil, then you need to clearly explain why neutrophils don't immediately self-activate.

PAR2 is a chemorepellent receptor on neutrophils, so you need to explain/ discuss why this transactivation doesn't result in neutrophil repulsion.

Minor points

Abstract

Deorphanized is pretty nonstandard, please replace this word

4 elicited \diamond activated

5 remission and grumbling (active) groups, grumbling is an unusual word.....

Fig 3A label total, cytosol, membrane

Throughout – the paper is a soup of acronyms, making it difficult to follow, decreasing acronyms would be immensely helpful

Reviewer #2 (Remarks to the Author):

Chu et al. report on the deorphanization of the adhesion receptor GPR97 as binding partner and allosteric activator of mPR3, which in turn activates PAR2 on human neutrophils. The authors present a plethora of biochemical data to elucidate the mechanism of action discussed. Among these methods, the authors engaged in structural analysis of the ectodomain/extracellular region of GPR97 (residues 1-264; abbrev. GRP97-ECR) by X-ray crystallography and present a structure at $\sim 3.4 \text{ \AA}$. Whereas the structural work is sound the presentation of it raises questions that need to be addressed.

Major points:

Page 8

It is not clear to the Reviewer why a positive F_0-F_c map of the size of an ion in a $\sim 3.4 \text{ \AA}$ electron density map should indicate the presence of a mixture of autoproteolysed and non-proteolysed forms within the crystal (Fig. 4C).

Could the authors please elaborate on this?

Although the autoproteolysis site is given in Fig. 2a and Suppl. Fig. 4e, it is not clear to the reader why the region showed in Fig. 4C should be of importance.

Could the authors please provide more information in the text?

In X-ray crystallography it is a fact that structural heterogeneity may lead to low resolution maps. In addition, from the biochemical point of view the authors showed that GRP97-ECR consist of an auto-cleavage site (Suppl. Fig. 4f).

Thus, it is not clear to the Reviewer why the authors did not try to circumvent this structural heterogeneity by mutating the autoproteolysis motif (residues 248-250) HLT to ALA as previously reported by Ping, YQ. et al, Nature 589, 620–626 (2021)?

The reviewer misses a clear statement, which residues could be resolved using the obtained map(s).

Could the authors please provide this information?

A second dataset, i.e. the S-SAD dataset, was recorded for correct annotation of the N-terminal region, but the dataset is not mentioned in Suppl. Table 4.

Could the authors please provide this information?

It is not clear to the Reviewer why the authors engage in a lengthy explanation and structural alignment (Fig. 4e) just to exclude the “density above the alpha-helix of the subdomain A” from subdomain A, since structural differences of individual subdomains A shown in Suppl. Fig. 4g/h are substantial.

Could the authors please provide this information?

From a structural point of view, it would be important to illustrate the “extensive hydrophobic interactions» between the alpha helices mentioned.

Thus, the Reviewer suggests moving Fig. 4d & 4e to the supplement but provide a panel displaying the “extensive hydrophobic interactions» of alpha helices.

Minor points:

The reviewer suggests deleting Fig. 4f since no gain in information was obtained in comparison to Fig. 4a.

Page 19

UniProt Q8R0T6 refers to AGRG3_MOUSE and not human GPR97. UniProt Q86Y34 would be the correct entry, isn't it?

A comparison of the experimental structure reported in this work with the alpha-fold model AF-Q86Y34-F1 would be interesting for the community as well as a brief discussed in the manuscript (e.g., experimental vs alpha-fold model).

Reviewer #3 (Remarks to the Author):

The GPCR PAR2 is a receptor that is activated through proteolysis by trypsin and other proteases, and can also be activated by transactivation through PAR1. This paper reports that transactivation of PAR2 by another GPCR, GPR97, in neutrophils, leading to stimulation of neutrophil responses.

The transactivation involves a complex of cell surface proteins including the protease mPR3, and CD177 and CD16b, in addition to PAR2 and GPR97. mPR3 is presented on the cell surface by CD177. CD177-associated mPR3 is identified as the ligand of GPR97, thus de-orphanising the receptor. A crystal structure of the mPR3 binding region of GPR97 is provided. The proposed mechanism is that binding of mPR3/CD177 to GPR97 activates mPR3, allowing it to cleave and thus activate PAR2.

The paper is very well written and the data are convincing. This is a substantial advance in mechanistic understanding of signals leading to neutrophil activation in the inflammatory response.

Major

1) One thing that remains unclear is how important this new mechanism of transactivation is for

neutrophil PAR2 activity compared to other mechanisms of PAR2 activation. Could the authors please compare transactivation by GPR97 with direct activation of PAR2 (e.g. by trypsin), for a couple of neutrophil responses, e.g. those shown in Fig 7A/B?

2) Statistics in Fig 7A,B,D,E and Supplem Fig 3B,D seem to have been done on normalised data. As far as I am aware, that is not permissible. The authors should analyse the raw data, if necessary with advice from a professional statistician.

Minor

3) $N\phi$: I don't like this abbreviation for 'neutrophils'. The word neutrophils should be spelled out in full. $N\phi$ is the symbol for magnetic flux linkage in physics.

4) It would be useful to define key abbreviations in the first figure legend, to help the reader (e.g. HC, MPA, GPA)

5) P7: 'both assays show....binds to PR3 directly and specifically': please rephrase to tone down conclusion of 'direct' binding from these data. The FACS-based assay does not demonstrate direct binding, and the 'ELISA-like' binding assay was done with proteins purified from HEK293 cells, so there remains some chance that contaminants contribute to or mediate the binding. The crystal structure provided later on in the manuscript provides convincing evidence of direct binding.

6) The fusion protein probe was used as the 'primary antibody'. Please rephrase. The fusion protein is not an antibody.

7) Fig 7A/B, the effect of the transactivation on bacterial killing is relatively minor compared to the effect on phagocytosis. Please comment on this discrepancy briefly in the discussion.

Response to reviewers' comments

The point by point answers to reviewers' comments are detailed below.

Reviewer #1 (Remarks to the Author):

This is a huge and well-developed study showing that a neutrophil surface receptor protein called GPR97 activates a protein complex on neutrophils, ultimately causing the activation of a different receptor, PAR2, and this in turn activates neutrophils. The report includes a crystal structure of the extracellular domain of GPR97, and a detailed molecular analysis of the interaction of this domain with the neutrophil cell surface protein complex. The writing is mostly clear, and the statistical analysis is good. The key thing missing from this report is the simple interpretation: what ultimately activates this mechanism, and why does this mechanism exist?

Authors: We thank the reviewer for the positive comments and the critical questions. As described in the revised manuscript and in the answers to the comments of Reviewers 1 and 3, we address the “what” and “why” questions briefly below:

Neutrophil activation usually involves a multi-step process that is induced in a temporal fashion, as neutrophils respond to a range of inflammatory stimuli that result in differential activation effects. In short, neutrophils adapt different activation statuses/phenotypes depending on the stimulants received in the inflammatory milieu. In fact, resting neutrophils are normally primed first by certain inflammatory irritants before being further activated by stronger stimuli. To achieve this complex response behaviour there are many layers of control, executed by different molecular players. This is to ensure a tight and timely control over when powerful, and potentially harmful, immune effector molecules are unleashed by neutrophils. How this works on the molecular level is not well understood.

Our results reveal one such set of molecular players, as we discover PR3/CD177/GPR97/PAR2/CD16b form a key interactome in neutrophil activation. In resting neutrophils, the cell-surface expression levels of GPR97 and PAR2 are very low, while the other members (PR3, CD177, and CD16b) are highly expressed (Fig. 6 and new Supplementary Fig. 7, 8). Mostly localized in the intracellular granules, significant GPR97 and PAR2 are translocated to the cell surface only when neutrophils are activated by certain stimulants (Fig. 6 and new Supplementary Fig. 7, 8). Low surface expression levels of GPR97 and PAR2 in resting neutrophils therefore limit the spontaneous cell self-activation by this mechanism. We have done additional experiments to search for conditions that could increase surface expression of GPR97 and PAR2 in neutrophils, and thereby trigger neutrophil activation of the pathway. Our data show that most inflammatory activator/cytokine including LPS, IL-8, IFN- γ ,

and fMLF don't affect the expression of GPR97 and PAR2 (new Supplementary Fig. 7b). We detected significantly increased PAR2, but not GPR97, expression only in neutrophils stimulated by IFN- γ and IFN- γ + fMLF in long-term (12 hr) culture (new Supplementary Fig. 7c). So far, the conditions in which both GPR97 and PAR2 are up-regulated significantly are neutrophil stimulation by degranulation stimulants of azurophilic granules and via the FcR-dependent activation mechanism (Fig. 6 and Supplementary Fig. 8). Given that this is a very new activation mechanism further specific triggers may be revealed in future studies.

We suggest the novel GPR97-PAR2 transactivation reaction is induced only by unique inflammatory triggers, such as FcR-mediated signalling, that up-regulate surface GPR97 and PAR2 expression to a significant level (the answer to "what" question). We believe the reason for having the PR3/CD177/GPR97/PAR2/CD16b interactome, consequently a GPR97-PAR2 transactivation mechanism, is to provide an extra layer of control that acts on primed/activated neutrophils. If activated, it allows for full neutrophil activation. Our results suggest that this is important for an effective anti-microbial response and immune effector function (the answer to "why" question).

Major points:

1. Page 6/ fig 2: GPR97 is expressed on the plasma membrane of neutrophils, and you show that the extracellular domain of GPR97 activates neutrophils, so you need to explain why two neutrophils touching each other won't activate each other. If on the other hand you are arguing that GPR97 on a neutrophil activates the same neutrophil, then you need to clearly explain why neutrophils don't immediately self-activate.

Authors: As discussed above, the reason why two neighbouring neutrophils won't activate each other and why there is no immediate self-activation of neutrophils is mainly because of the very low surface levels of GPR97 and PAR2 in resting neutrophils. As such, no sufficient PR3/CD177/GPR97/PAR2/CD16b interactome is formed and hence no GPR97-mediated PAR2 transactivation. By contrast, uniquely primed/activated neutrophils upregulate GPR97 and PAR2 expression to permit the clustering and formation of the PR3/CD177/GPR97/PAR2/CD16b interactome, eventually leading to GPR97-PAR2 transactivation and further inflammatory activation. Our data support the idea that the PR3/CD177/GPR97/PAR2/CD16b interactome is clustered in *cis* on the membrane of the same neutrophil, however we can't completely rule out the possibility of the interactome in the *trans* configuration, ie the formation of receptor interactome via the close contact of two neighbouring primed/activated neutrophils.

2. PAR2 is a chemorepellent receptor on neutrophils, so you need to explain/ discuss why this transactivation doesn't result in neutrophil repulsion.

Authors: We thank the reviewer for pointing out the fact that PAR2 was identified as a neutrophil chemorepellent receptor. Unfortunately, due to the extensive works involved we did not investigate the effect of GPR97-PAR2 transactivation in neutrophil repulsion in the present manuscript. Nevertheless, our ongoing study has indicated a potential role of GPR97-PAR2 transactivation in the reverse transendothelial migration (rTEM) of neutrophils, supporting in part the role of PAR2 as a chemorepellent receptor of neutrophils. We believe this interesting question is a completely new research avenue and warrants a full and carefully-designed study, which is beyond the scope of the present report.

Minor points:

1. Abstract. Deorphanized is pretty nonstandard, please replace this word

Authors: As suggested, we have replaced it with "demonstrated" in the Abstract.

2. elicited activated

Authors: We have replaced "elicited" with "induced" or "activated" in the text.

3. remission and grumbling (active) groups, grumbling is an unusual word.....

Authors: We have removed "grumbling" and used "active" instead.

4. Fig 3A label total, cytosol, membrane

Authors: As suggested, we have labelled western blots in Fig. 3a with total lysate, cytosolic fraction, and membrane fraction.

5. Throughout – the paper is a soup of acronyms, making it difficult to follow, decreasing acronyms would be immensely helpful.

Authors: We thank the reviewer for the suggestion and we have removed several acronyms (7TM, Ag, GPI, APS, TIN ϕ s, and PI-PLC) from the text.

Reviewer #2 (Remarks to the Author):

Chu et al. report on the deorphanization of the adhesion receptor GPR97 as binding partner and allosteric activator of mPR3, which in turn activates PAR2 on human neutrophils. The authors present a plethora of biochemical data to elucidate the mechanism of action discussed. Among these methods, the authors engaged in structural analysis of the ectodomain/extracellular region of GPR97 (residues 1-264;

abbrev. GRP97-ECR) by X-ray crystallography and present a structure at ~ 3.4 Å. Whereas the structural work is sound the presentation of it raises questions that need to be addressed.

Authors: We thank the reviewer for the positive assessment of the structural work and address the remaining concerns below.

Major points:

1. Page 8. It is not clear to the Reviewer why a positive Fo-Fc map of the size of an ion in a ~ 3.4 Å electron density map should indicate the presence of a mixture of autoproteolysed and non-proteolysed forms within the crystal (Fig. 4C).

Could the authors please elaborate on this?

Authors: We thank the reviewer for this observation. The sample we subjected to crystallisation contained a mixture of autoproteolytically cleaved and non-cleaved forms of GPR97, which we were unable to separate from each other during protein purification. The electron density map suggests that a small fraction of uncleaved protein may still be present in the crystals, while most of the protein is cleaved. We have now amended the manuscript to remove any confusion on this (see below).

2. Although the autoproteolysis site is given in Fig. 2a and Suppl. Fig. 4e, it is not clear to the reader why the region showed in Fig. 4C should be of importance. Could the authors please provide more information in the text?

Authors: Indeed, we agree that the autoproteolytic site structure does not have a direct impact on the functional aspects of the study. It is a conserved feature of most GAIN domains, and we were pleased that it agrees with the two bands we observe on SDS-page (Suppl. Fig. 4a). We agree that it is not necessary to show the structure in detail and have therefore removed Fig. 4c. The main text was also updated to: “The electron density map also confirmed that the GPR97 GAIN domain is mostly autoproteolysed at its GPS motif, as expected and observed for other GAIN domains”.

3. In X-ray crystallography it is a fact that structural heterogeneity may lead to low resolution maps. In addition, from the biochemical point of view the authors showed that GRP97-ECR consist of an auto-cleavage site (Suppl. Fig. 4f).

Thus, it is not clear to the Reviewer why the authors did not try to circumvent this structural heterogeneity by mutating the autoproteolysis motif (residues 248-250) HLT to ALA as previously reported by Ping, YQ. et al, Nature 589, 620–626 (2021)?

Authors: We thank the reviewer for this suggestion. Our dataset was obtained in 2018, well before this new study on GPR97 was published. The then available structures of GAIN domains (4DLQ, 4DLO, 5KVM, 6V55) all used wild type sequences and showed a

cleaved but intact receptor. There are some advantages to using wild type protein whenever possible, for example to avoid artefacts due to mutations..

4. The reviewer misses a clear statement, which residues could be resolved using the obtained map(s). Could the authors please provide this information?

Authors: We thank the reviewer for this observation and have added this information in the main text as well: “we determined its atomic structure (residues 28-260) at 3.37 Å resolution using X-ray crystallography”.

5. A second dataset, i.e. the S-SAD dataset, was recorded for correct annotation of the N-terminal region, but the dataset is not mentioned in Suppl. Table 4. Could the authors please provide this information?

Authors: The information has been added to the Supplementary Table 4.

6. It is not clear to the Reviewer why the authors engage in a lengthy explanation and structural alignment (Fig. 4e) just to exclude the “density above the alpha-helix of the subdomain A” from subdomain A, since structural differences of individual subdomains A shown in Suppl. Fig. 4g/h are substantial. Could the authors please provide this information?

Authors: We thank the reviewer for this comment and have removed the confusing phrasing.

7. From a structural point of view, it would be important to illustrate the “extensive hydrophobic interactions» between the alpha helices mentioned. Thus, the Reviewer suggests moving Fig. 4d & 4e to the supplement but provide a panel displaying the “extensive hydrophobic interactions» of alpha helices.

Authors: We agree and have provided a panel displaying the extensive hydrophobic interactions between the NTD and GAIN domains in Figure 4c.

Minor points:

1. The reviewer suggests deleting Fig. 4f since no gain in information was obtained in comparison to Fig. 4a.

Authors: We have updated the figure accordingly.

2. Page 19. UniProt Q8ROT6 refers to AGRG3_MOUSE and not human GPR97. UniProt Q86Y34 would be the correct entry, isn't it?

Authors: We thank the reviewer for spotting this mistake and apologise for this. The text has been updated with the correct accession number.

3. A comparison of the experimental structure reported in this work with the alpha-fold model AF-Q86Y34-F1 would be interesting for the community as well as a brief discussed in the manuscript (e.g., experimental vs alpha-fold model).

Authors: We thank the reviewer for this suggestion and have included a brief comparison in the main text as well as modified the supplementary figure. The updated text reads as “Structural comparison with a model calculated by alpha-fold³⁰ shows that the GAIN domain is similar, but the NTD-helix is off-set compared to our experimental model. Alpha-fold predicts residues Q21-G27 to be flexible (Supplementary Fig. 4i)”.

Reviewer #3 (Remarks to the Author):

The GPCR PAR2 is a receptor that is activated through proteolysis by trypsin and other proteases, and can also be activated by transactivation through PAR1. This paper reports that transactivation of PAR2 by another GPCR, GPR97, in neutrophils, leading to stimulation of neutrophil responses.

The transactivation involves a complex of cell surface proteins including the protease mPR3, and CD177 and CD16b, in addition to PAR2 and GPR97. mPR3 is presented on the cell surface by CD177. CD177-associated mPR3 is identified as the ligand of GPR97, thus de-orphanising the receptor. A crystal structure of the mPR3 binding region of GPR97 is provided. The proposed mechanism is that binding of mPR3/CD177 to GPR97 activates mPR3, allowing it to cleave and thus activate PAR2.

The paper is very well written and the data are convincing. This is a substantial advance in mechanistic understanding of signals leading to neutrophil activation in the inflammatory response.

Authors: We appreciate the positive assessment of our work by the reviewer. Our answers to the comments are listed below.

Major points:

1) One thing that remains unclear is how important this new mechanism of transactivation is for neutrophil PAR2 activity compared to other mechanisms of PAR2 activation. Could the authors please compare transactivation by GPR97 with direct activation of PAR2 (e.g. by trypsin), for a couple of neutrophil responses, e.g. those shown in Fig 7A/B?

Authors: We thank the reviewer for the suggestion. We have performed new experiments to directly compare neutrophil activation phenotypes induced by PAR2-specific activator (trypsin) and agonistic peptides (SLIGRL-NH₂ and SLIGKV-NH₂) with those induced by GPR97^E-mFc (new Supplementary Fig. 9). The phenotypes examined include morphological changes, IL-8 and ROS production, and bacteria uptake and killing. In comparison to GPR97^E-mFc stimulation, trypsin treatment induced fewer morphological changes in neutrophils, and no morphological changes were induced by the SLIGRL-NH₂ and SLIGKV-NH₂ peptides (Supplementary Fig. 9a). In the serum-free culture condition (due to the use of trypsin), IL-8 production was comparably enhanced in neutrophils treated with GPR97^E-mFc versus the SLIGRL-NH₂ and SLIGKV-NH₂ peptides. However, the concentrations of IL-8 secreted in this condition were very low, most likely because of the serum-free medium. Interestingly, no increased IL-8 production was found in trypsin-treated neutrophils (Supplementary Fig. 9b). By contrast, in the standard culture condition using RPMI complete medium (with 10% FCS), only GPR97^E-mFc treated neutrophils produced significantly increased IL-8 (Supplementary Fig. 9c).

Likewise, ROS production and bacteria uptake/killing abilities done in the standard experimental condition were enhanced mostly in neutrophils treated with GPR97^E-mFc (Supplementary Fig. 9d-f). Of note, neutrophils treated with the SLIGRL-NH₂ agonistic peptide but not the SLIGKV-NH₂ peptide showed a similar *E.coli* uptake and killing ability in comparison to those incubated with GPR97^E-mFc (Supplementary Fig. 9e). In contrast, no obvious effects were identified on *S. typhimurium* uptake and killing in neutrophils incubated with SLIGRL-NH₂ and SLIGKV-NH₂ (Supplementary Fig. 9e). The reason for the differential effects of SLIGRL-NH₂ (mouse PAR2 agonist) and SLIGKV-NH₂ (human PAR2 agonist) peptides on the *E.coli* uptake and killing abilities of neutrophils is unknown. Hence, while PAR2-specific activator and agonistic peptides do activate PAR2 in resting neutrophils, the activation phenotypes induced are relatively mild in comparison to those induced by GPR97^E-mFc. These results suggest that the GPR97-PAR2 transactivation mechanism likely represents one of the predominant triggers of PAR2 activation in neutrophils.

2) Statistics in Fig 7A,B,D,E and Suppl Fig 3B,D seem to have been done on normalised data. As far as I am aware, that is not permissible. The authors should analyse the raw data, if necessary with advice from a professional statistician.

Authors: We thank the reviewer for the critical questions. We agree with the reviewer that it is atypical to use normalized data in most studies. Nevertheless, due to the fact that the percentage of CD177⁺ (also mPR3⁺) neutrophils, the target cell population of

the present study, varies widely (0-100%) in normal populations, the data generated using neutrophils from different individuals tended to diverge significantly as well. This is especially true in the analysis of biological functions such as bacteria uptake/killing and endothelial permeability, even though the individual sets of data all showed a very similar trend. It was thus quite difficult to compare the raw data of many independent experiments directly. Consequently, we chose instead to use normalised data in the form of relative fold changes over negative controls. Due to the unique nature of highly variable CD177 expression profiles in human neutrophils, we believe the use of normalised data representing the results of these functional assays is reasonable. Similar approaches are being taken by other authors in the field, e.g., Bai *et al.* CD177 modulates human neutrophil migration through activation-mediated integrin and chemoreceptor regulation (Blood 2017; 130:2092-2100. Figures 1, 2, 5 and 6).

Minor points:

3) N ϕ : I don't like this abbreviation for 'neutrophils'. The word neutrophils should be spelled out in full. N ϕ is the symbol for magnetic flux linkage in physics.

Authors: As requested, we have spelled out neutrophil in full in the main text, figure legends, and figures.

4) It would be useful to define key abbreviations in the first figure legend, to help the reader (e.g. HC, MPA, GPA)

Authors: As suggested, we have defined the abbreviations (HC, MPA, GPA) in the legends of Fig. 1.

5) P7: 'both assays show....binds to PR3 directly and specifically': please rephrase to tone down conclusion of 'direct' binding from these data. The FACS-based assay does not demonstrate direct binding, and the 'ELISA-like' binding assay was done with proteins purified from HEK293 cells, so there remains some chance that contaminants contribute to or mediate the binding. The crystal structure provided later on in the manuscript provides convincing evidence of direct binding.

Authors: As suggested, we have removed "directly" from the sentence in page 7.

6) The fusion protein probe was used as the 'primary antibody'. Please rephrase. The fusion protein is not an antibody.

Authors: As suggested, we have removed "as the primary Ab" from the sentence in page 18.

7) Fig 7A/B, the effect of the transactivation on bacterial killing is relatively minor compared to the effect on phagocytosis. Please comment on this discrepancy briefly in the discussion.

Authors: As suggested, we have added a sentence in the Discussion section in page 13 to describe these differential effects.

REVIEWERS' COMMENTS

Reviewer #1 (Remarks to the Author):

Thank you fo the nice job answering my questions. My only suggestion is to put some of what you wrote so nicely in the reply to my main question ("...we address the "what" and "why" questions briefly below...") in the text.

Reviewer #2 (Remarks to the Author):

The Reviewer thanks the authors for the revised version and the point-by-point discussion.

My points have been addressed satisfactory - congratulations to the authors.

Reviewer #3 (Remarks to the Author):

The authors have addressed all my comments in the revised mansucript. They put considerable effort into this revision by performing a number of additinal experiments to evaluate the importance of the novel pathway they identified compared to previously known mechanisms of activation of this receptor. I am very pleased that the novel pathway is even more important than the previously known mechanisms.

Response to reviewers' comments

The point by point answers to reviewers' comments are detailed below.

Reviewer #1 (Remarks to the Author):

Thank you for the nice job answering my questions. My only suggestion is to put some of what you wrote so nicely in the reply to my main question ("...we address the "what" and "why" questions briefly below...") in the text.

Authors: We thank the reviewer for the positive comments and the suggestion. As requested, we have added a paragraph at the end of Discussion section (p15) to emphasize the "what" and "why" role of GPR97-PAR2 activation in the inflammatory functions of neutrophil leukocytes.

Reviewer #2 (Remarks to the Author):

The Reviewer thanks the authors for the revised version and the point-by-point discussion.

My points have been addressed satisfactory - congratulations to the authors.

Authors: We thank the reviewer for the positive comment.

Reviewer #3 (Remarks to the Author):

The authors have addressed all my comments in the revised manuscript. They put considerable effort into this revision by performing a number of additional experiments to evaluate the importance of the novel pathway they identified compared to previously known mechanisms of activation of this receptor. I am very pleased that the novel pathway is even more important than the previously known mechanisms.

Authors: We thank the reviewer for the nice comment.